# Postoperative mortality analysis on nationwide data from diagnosis procedure combination database in Japan

**Susumu Kunisawa**\* 

Department of Healthcare Economics and Quality Management, Graduate School of Medicine, Kyoto University, Kyoto City, Japan

\* kunisawa.susumu.2v@kyoto-u.ac.jp

## Abstract

### Introduction

The present study aimed to investigate the postoperative mortality due to all surgeries at the prefectural level using a nationwide diagnosis procedure combination (DPC) database in Japan and to evaluate the data according to temporal changes and regional differences.

### Methods

Data were provided in accordance with the guidelines indicated on the Ministry of Health, Labor and Welfare, Japan. The number of cases and in-hospital mortality were calculated for each representative surgery for each hospitalization according to fiscal year of discharge from 2011 to 2018 and according to prefecture. Values of $\geq 10$ in each aggregated data cell were presented.

### Results and discussion

The aggregated result data contain 474,154 records, with about 2,000 different surgical codes. More than 10 mortalities were recorded in only 16,890 data cells, which can be used in the mortality analysis. In the analyses of artificial head insertion, cerebral aneurysm neck clipping, coronary artery and aortic bypass grafting, and tracheotomy, regional differences and a declining trend were observed in some categories.

### Conclusion

In addition to considering categories that can be used in the analysis, careful consideration must be given to the inclusion of background context such as the quality of care.

## 1. Introduction

Improving quality of care is one of the greatest concerns in the medical community, and reducing postoperative complications deserves attention. This is a common concern worldwide; however, there have been reports of relatively few perioperative complications in Japan compared to other countries [1, 2]. In Japan, various case registries have been conducted, verified, and

**Data Availability Statement:** All relevant data are within the manuscript and its Supporting information files.

**Funding:** This study was supported by Japan Society for the Promotion of Science (grant

number 20K18961). The funders had no role in study design, data collection and analysis, decision to publish, or preparation of the manuscript.

**Competing interests:** The authors have declared that no competing interests exist.

reviewed, mainly at academic conferences [3, 4]. Currently, a large volume of case data is available the Japanese surgical field, primarily through the National Clinical Database (NCD), and these data are being used for validation and feedback to facilities. However, data that can be obtained from such case registries is limited to the data registered in them. Although some surgical procedures have fairly high registration rates [5], registration must be a proactive approach; moreover, registration omissions can still occur. In addition, databases, such as the NCD (https://www.ncd.or.jp/press/), do not necessarily include data from all surgical areas, although they are being used in major surgeries such as gastroenterological surgery, cardiovascular surgery, respiratory surgery, and pediatric surgery [6–8]. This can be a limiting factor for studies based on any case registries that are conducted not only in Japan but also in other countries.

Exhaustive analyses using reimbursement or administrative data that are not registry databases are readily being conducted. In Japan, along with the evolution of information technology, the diagnosis procedure combination (DPC) database that included data obtained via the DPC per-diem payment system (DPC/PDPS), which was started in 2003, was created in tandem with the accumulation and utilization of DPC data. DPC data is prepared as a discharge summary called "Form 1" in addition to documenting—at the voucher level—the medical treatment activities performed during treatment, despite being a comprehensive payment system. Initially, these data were defined according to the DPC/PDPS system; however, because medical institutions that have not adopted DPC/PDPS were also approached for creating the DPC database, this database now includes data created and submitted to the Ministry of Health, Labor and Welfare (MHLW) by the majority of hospitals (5,315 hospitals in 2020; https://www.mhlw.go.jp/stf/shingi2/0000196043_00005.html).

The DPC database, containing data on most hospitalizations related to insured care, has been applied in practical management and clinical research analyses from various perspectives. This database has been very useful for research across organizations because it contains government-defined data. Notably, many intrahospital analyses and cross-sectional or interhospital analyses within multiple organizations or groups have been conducted [9–13]. Moreover, since 2017, data collected by the MHLW could be used for analyses. Previously, a study successfully calculated process indicators using this database [14] and distinctly showed the temporal changes and regional differences in Japan.

Postoperative mortality can be considered one of the quality indicators. Although mortality is associated with various parameters such as the technical quality, case-mix differences, and differential indication strategies, the mortality rates in Japan are often lower than those in other countries [1, 2]. However, such results are obtained from the analysis of data from specific hospitals or from a data registry. Therefore, the present study was designed to explore regional differences and temporal changes in postoperative mortality for any surgery, with quality of care as an outcome measure using a comprehensive nationwide database from Japan. Instead of examining details such as the causes of regional differences in specific surgeries, the present study fundamentally calculated and provided these indicators, thereby providing further insights and facilitating detailed and specific studies in the future. In this analysis, some stipulations for data usage for cross-disciplinary indicators are presented and several items that should be considered by future investigators are reported.

## 2. Material and methods

Data were obtained in accordance with the guidelines and applications indicated on the MHLW website. The number of cases and in-hospital mortality were calculated for each representative surgery for each hospitalization according to the fiscal year of discharge from 2011 to 2018 and according to prefecture; data were provided as an aggregated result from the

MHLW. The "Form 1" of DPC data facilitates the recording of a maximum of five surgeries, and the primary surgery or the one with the highest surgical fee is documented as the first surgery. Surgical information recorded included the surgery name and surgical code (K-code). In this study, data were analyzed and presented based on K-codes. As this coding is specifically used only in the Japanese healthcare system, the used coding system has been provided in the S1 Data. As the surgery name causes some notational inconsistencies, data were tabulated according to the surgical code; the standard master surgery names for those codes in Japanese are available at the Various Information of Medical Fee website (http://shinryohoshu.mhlw.go.jp/shinryohoshu/), and the referable translation is available at the Japan Surgical Society—Web Glossary of Surgical Terms (http://yougosyu.jssoc.or.jp/). Data that can be obtained as an aggregate has been limited by the rules of MHLW to a value of ≥10 in each cell for anonymity. If the obtained value was <10, it was removed and nullified, but was indicated as "-" in this report. Not all of these obtained aggregated data are disclosed in this report in accordance with the guidelines of MHLW; in addition, the number was removed and indicated as "*-" when the difference between the number of cases and in-hospital mortality was <10 (in S1 Data). In the present study, postoperative in-hospital mortality was defined and calculated as the number of in-hospital mortality divided by the number of postoperative case discharges. Therefore, patients who were discharged alive once and died during events such as readmission were not included in the mortality data.

The number of DPC-participating hospitals is increasing annually. As mentioned above, the submission of DPC data has been expanded to non-DPC hospitals in recent years, with each progressive year showing an increase in the number of hospitals submitting DPC data. Consequently, more cases are being covered by this database; thus, the case volume has increased.

All surgical data obtained in the present study is presented as S1 Data. Across the entire spectrum, each result has its own importance and requires interpretation, but this report selected the most distinctive ones for consideration. Although this selection was arbitrarily made by the author, the inclusion criteria were as follows: those with relatively high mortality rates, those with visible temporal changes or regional differences, those requiring interpretation, or those requiring caution in future data analysis.

The present study was approved by Kyoto University Graduate School and Faculty of Medicine, Ethics Committee. The need for obtaining individual informed consent was waived for this study as that data were anonymized and provided by the MHLW in accordance with the "Strategy for the Revitalization of Japan" 2016 (approved by the Cabinet on June 2, 2016). The data presented in this study were independently requested and analyzed by the author and are different from the data such as statistics prepared and published by the MHLW Japan.

## 3. Results

All data obtained in this study are presented as S1 Data, except for the unknown operation code data of 24 records. The dataset contained 474,154 records; for each year, the dataset comprised 1,986–2,496 different codes. Records with 0 cases were not included in the dataset; 221,866 records contained >10 cases. Only 16,890 records exhibited >10 mortalities, and 186 records masked the methods. With 56 surgical codes, the mortality rates for all prefectures during a fiscal year were comparable, and with 1,063 surgical codes, the mortality rates during a fiscal year were comparable among prefectures.

### 3.1. Artificial head insertion for the shoulder or hip (K0811)

Fig 1 depicts the postoperative in-hospital mortality of artificial head insertion for the shoulder or hip in orthopedic surgery, and Table 1 shows the detail results for those cases.

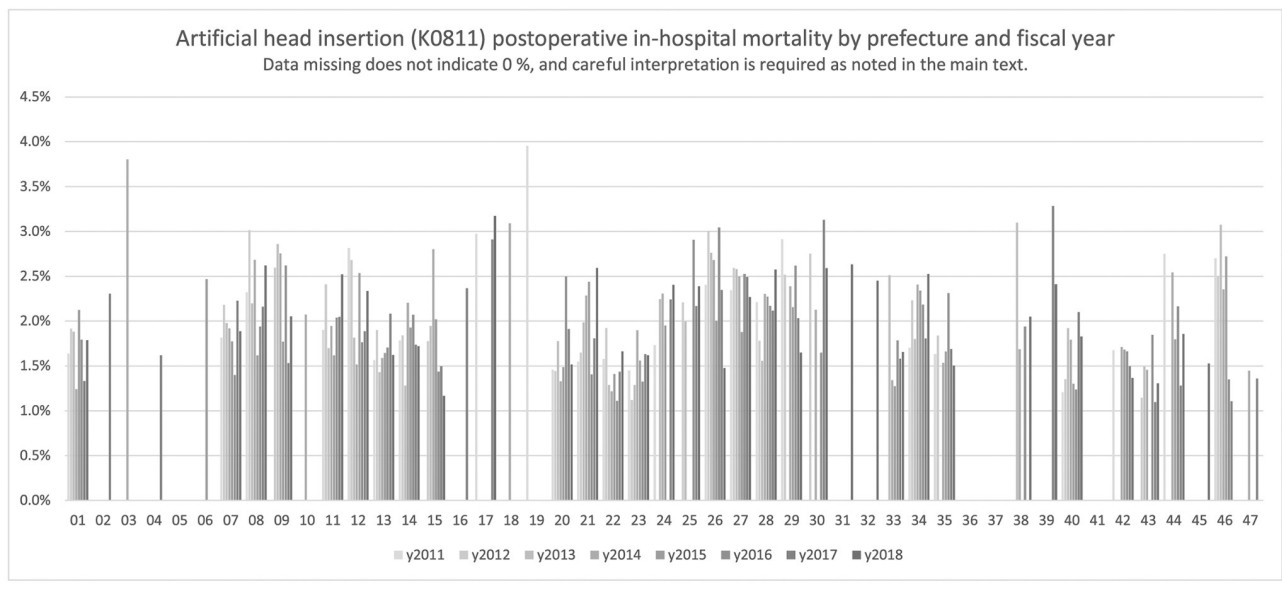

**Fig 1. Artificial head insertion for the shoulder or hip (K0811).** Artificial head insertion (K0811) postoperative in-hospital mortality by prefecture and fiscal year.

Surgeries coded with K0811 were artificial head insertions for the shoulder or hip; therefore, the cases were mixed. Moreover, postoperative in-hospital mortality rate was approximately 1%–3% and appeared to slightly vary regionally. However, there were many regions where numerical aggregation was impossible because of low postoperative in-hospital mortalities in each area.

### 3.2. Cerebral aneurysm neck clipping (1 location; K1771)

Fig 2 depicts the postoperative in-hospital mortality of 1 location cerebral aneurysm neck clipping, and Table 2 shows the results for those cases. The code K1771 and K1772 were used to indicate clipping of one location and multiple locations, respectively. A decreasing trend over time can be observed in 14 (Miyagi Prefecture), 22 (Shizuoka Prefecture), and 23 (Aichi Prefecture) records.

### 3.3. Coronary artery and aortic bypass grafting (K5521, K5522, K552-21, and K552-22)

Fig 3 depicts the postoperative in-hospital mortality in coronary artery and aortic bypass grafting of >2 anastomoses (K5522, Table 4), and Tables 3–6 show the results for coronary artery and aortic bypass grafting with 1 anastomosis using a heart–lung machine (K5521), >2 anastomoses using a heart–lung machine (K5522), with 1 anastomosis without a heart–lung machine (K552-21), and >2 anastomoses without a heart–lung machine (K552-22). Data shown in Table 4 included relatively more numerical results; for data shown the remaining tables, numerical results were available for the denominator, and the number of numerators in several regions was too small to be tabulated. Although only a few areas can be depicted as a result of the tally, regional differences were noted. As mentioned above, coronary artery bypass surgery was classified into four reimbursement categories. These could be considered different

**Table 1. Artificial head insertion (K0811): Case number and outcomes of postoperative in-hospital mortality.**

| Fiscal year | 2011 | | | 2012 | | | 2013 | | | 2014 | | | 2015 | | | 2016 | | | 2017 | | | 2018 | | |
|---|---|---|---|---|---|---|---|---|---|---|---|---|---|---|---|---|---|---|---|---|---|---|---|---|
| Prefecture | Case | Outcome | | Case | Outcome | | Case | Outcome | | Case | Outcome | | Case | Outcome | | Case | Outcome | | Case | Outcome | | Case | Outcome | |
| 01 | 1221 | 20 | 1.6% | 1358 | 26 | 1.9% | 1436 | 27 | 1.9% | 1692 | 21 | 1.2% | 1978 | 42 | 2.1% | 2119 | 38 | 1.8% | 2254 | 30 | 1.3% | 2406 | 43 | 1.8% |
| 02 | 286 | - | - | 321 | - | - | 286 | - | - | 342 | - | - | 378 | - | - | 432 | - | - | 477 | 11 | 2.3% | 500 | - | - |
| 03 | 319 | - | - | 334 | - | - | 329 | - | - | 368 | 14 | 3.8% | 399 | - | - | 364 | - | - | 453 | - | - | 528 | - | - |
| 04 | 555 | - | - | 536 | - | - | 486 | - | - | 619 | - | - | 613 | - | - | 744 | - | - | 741 | 12 | 1.6% | 816 | - | - |
| 05 | 306 | - | - | 265 | - | - | 290 | - | - | 312 | - | - | 357 | - | - | 352 | - | - | 385 | - | - | 371 | - | - |
| 06 | 390 | - | - | 375 | - | - | 363 | - | - | 379 | - | - | 405 | 10 | 2.5% | 378 | - | - | 452 | - | - | 435 | - | - |
| 07 | 551 | 10 | 1.8% | 550 | 12 | 2.2% | 556 | 11 | 2.0% | 626 | 12 | 1.9% | 676 | 12 | 1.8% | 715 | 10 | 1.4% | 808 | 18 | 2.2% | 849 | 16 | 1.9% |
| 08 | 775 | 18 | 2.3% | 796 | 24 | 3.0% | 683 | 15 | 2.2% | 895 | 24 | 2.7% | 928 | 15 | 1.6% | 1032 | 20 | 1.9% | 1204 | 26 | 2.2% | 1221 | 32 | 2.6% |
| 09 | 366 | - | - | 424 | 11 | 2.6% | 490 | 14 | 2.9% | 581 | 16 | 2.8% | 677 | 12 | 1.8% | 687 | 18 | 2.6% | 784 | 12 | 1.5% | 877 | 18 | 2.1% |
| 10 | 411 | - | - | 499 | - | - | 498 | - | - | 579 | 12 | 2.1% | 664 | - | - | 702 | - | - | 763 | - | - | 691 | - | - |
| 11 | 1420 | 27 | 1.9% | 1535 | 37 | 2.4% | 1648 | 28 | 1.7% | 1954 | 38 | 1.9% | 2289 | 37 | 1.6% | 2651 | 54 | 2.0% | 2834 | 58 | 2.0% | 3094 | 78 | 2.5% |
| 12 | 1173 | 33 | 2.8% | 1269 | 34 | 2.7% | 1321 | 24 | 1.8% | 1650 | 25 | 1.5% | 1855 | 47 | 2.5% | 2267 | 40 | 1.8% | 2334 | 44 | 1.9% | 2526 | 59 | 2.3% |
| 13 | 3709 | 58 | 1.6% | 3843 | 73 | 1.9% | 4198 | 60 | 1.4% | 4720 | 75 | 1.6% | 5049 | 83 | 1.6% | 5341 | 91 | 1.7% | 5957 | 124 | 2.1% | 6230 | 101 | 1.6% |
| 14 | 2860 | 51 | 1.8% | 3047 | 56 | 1.8% | 3358 | 43 | 1.3% | 3721 | 82 | 2.2% | 4050 | 78 | 1.9% | 4300 | 89 | 2.1% | 4547 | 79 | 1.7% | 4706 | 81 | 1.7% |
| 15 | 547 | - | - | 619 | 11 | 1.8% | 720 | 14 | 1.9% | 785 | 22 | 2.8% | 990 | 20 | 2.0% | 975 | 14 | 1.4% | 1136 | 17 | 1.5% | 1117 | 13 | 1.2% |
| 16 | 407 | - | - | 412 | - | - | 459 | - | - | 433 | - | - | 564 | - | - | 540 | - | - | 549 | 13 | 2.4% | 516 | - | - |
| 17 | 370 | 11 | 3.0% | 350 | - | - | 331 | - | - | 404 | - | - | 446 | - | - | 434 | - | - | 481 | 14 | 2.9% | 504 | 16 | 3.2% |
| 18 | 297 | - | - | 339 | - | - | 342 | - | - | 356 | 11 | 3.1% | 315 | - | - | 327 | - | - | 344 | - | - | 381 | - | - |
| 19 | 253 | 10 | 4.0% | 267 | - | - | 275 | - | - | 282 | - | - | 306 | - | - | 365 | - | - | 425 | - | - | 469 | - | - |
| 20 | 686 | 10 | 1.5% | 763 | 11 | 1.4% | 788 | 14 | 1.8% | 904 | 12 | 1.3% | 943 | 14 | 1.5% | 960 | 24 | 2.5% | 1047 | 20 | 1.9% | 1056 | 16 | 1.5% |
| 21 | 905 | 14 | 1.5% | 909 | 15 | 1.7% | 857 | 17 | 2.0% | 1050 | 24 | 2.3% | 1107 | 27 | 2.4% | 1068 | 15 | 1.4% | 1217 | 22 | 1.8% | 1235 | 32 | 2.6% |
| 22 | 1522 | 24 | 1.6% | 1509 | 29 | 1.9% | 1555 | 20 | 1.3% | 1727 | 21 | 1.2% | 1774 | 25 | 1.4% | 1891 | 21 | 1.1% | 2022 | 29 | 1.4% | 1985 | 33 | 1.7% |
| 23 | 1796 | 26 | 1.4% | 2052 | 23 | 1.1% | 2254 | 29 | 1.3% | 2424 | 46 | 1.9% | 2758 | 43 | 1.6% | 2872 | 38 | 1.3% | 3006 | 49 | 1.6% | 3024 | 49 | 1.6% |
| 24 | 693 | 12 | 1.7% | 704 | - | - | 757 | 17 | 2.2% | 823 | 19 | 2.3% | 923 | 18 | 2.0% | 975 | - | - | 1160 | 26 | 2.2% | 1123 | 27 | 2.4% |
| 25 | 462 | - | - | 453 | 10 | 2.2% | 501 | 10 | 2.0% | 526 | - | - | 598 | - | - | 654 | 19 | 2.9% | 692 | 15 | 2.2% | 670 | 16 | 2.4% |
| 26 | 790 | 19 | 2.4% | 965 | 29 | 3.0% | 978 | 27 | 2.8% | 1082 | 29 | 2.7% | 1198 | 24 | 2.0% | 1215 | 37 | 3.0% | 1364 | 32 | 2.3% | 1423 | 21 | 1.5% |
| 27 | 2986 | 70 | 2.3% | 3280 | 85 | 2.6% | 3376 | 87 | 2.6% | 3961 | 99 | 2.5% | 4426 | 83 | 1.9% | 4593 | 116 | 2.5% | 4973 | 124 | 2.5% | 5376 | 122 | 2.3% |
| 28 | 1764 | 39 | 2.2% | 1910 | 34 | 1.8% | 1991 | 31 | 1.6% | 2302 | 53 | 2.3% | 2815 | 64 | 2.3% | 2996 | 65 | 2.2% | 3119 | 66 | 2.1% | 3225 | 83 | 2.6% |
| 29 | 618 | 18 | 2.9% | 596 | 15 | 2.5% | 685 | - | - | 628 | 15 | 2.4% | 742 | 16 | 2.2% | 764 | 20 | 2.6% | 837 | 17 | 2.0% | 1031 | 17 | 1.6% |
| 30 | 489 | - | - | 545 | 15 | 2.8% | 523 | - | - | 612 | 13 | 2.1% | 610 | - | - | 668 | 11 | 1.6% | 767 | 24 | 3.1% | 734 | 19 | 2.6% |
| 31 | 234 | - | - | 305 | - | - | 266 | - | - | 295 | - | - | 403 | - | - | 382 | - | - | 370 | - | - | 418 | 11 | 2.6% |
| 32 | 240 | - | - | 313 | - | - | 293 | - | - | 349 | - | - | 371 | - | - | 393 | - | - | 465 | - | - | 408 | 10 | 2.5% |
| 33 | 681 | - | - | 805 | - | - | 796 | 20 | 2.5% | 896 | 12 | 1.3% | 1021 | 13 | 1.3% | 1064 | 19 | 1.8% | 1202 | 19 | 1.6% | 1147 | 19 | 1.7% |
| 34 | 998 | 17 | 1.7% | 1076 | 24 | 2.2% | 1112 | 20 | 1.8% | 1246 | 30 | 2.4% | 1368 | 32 | 2.3% | 1511 | 33 | 2.2% | 1606 | 29 | 1.8% | 1743 | 44 | 2.5% |
| 35 | 612 | 10 | 1.6% | 653 | 12 | 1.8% | 680 | - | - | 782 | 12 | 1.5% | 843 | 14 | 1.7% | 865 | 20 | 2.3% | 947 | 16 | 1.7% | 997 | 15 | 1.5% |
| 36 | 322 | - | - | 411 | - | - | 405 | - | - | 524 | - | - | 554 | - | - | 621 | - | - | 587 | - | - | 592 | - | - |
| 37 | 390 | - | - | 411 | - | - | 430 | - | - | 432 | - | - | 463 | - | - | 466 | - | - | 559 | - | - | 557 | - | - |
| 38 | 355 | - | - | 396 | - | - | 452 | 14 | 3.1% | 593 | 10 | 1.7% | 687 | - | - | 722 | 14 | 1.9% | 794 | - | - | 878 | 18 | 2.1% |
| 39 | 329 | - | - | 353 | - | - | 356 | - | - | 369 | - | - | 388 | - | - | 369 | - | - | 457 | 15 | 3.3% | 456 | 11 | 2.4% |
| 40 | 1991 | 24 | 1.2% | 2071 | 28 | 1.4% | 2136 | 41 | 1.9% | 2344 | 42 | 1.8% | 2537 | 33 | 1.3% | 2750 | 34 | 1.2% | 2906 | 61 | 2.1% | 2953 | 54 | 1.8% |
| 41 | 320 | - | - | 334 | - | - | 341 | - | - | 389 | - | - | 485 | - | - | 481 | - | - | 498 | - | - | 473 | - | - |
| 42 | 657 | 11 | 1.7% | 674 | - | - | 701 | - | - | 818 | 14 | 1.7% | 833 | 14 | 1.7% | 902 | 15 | 1.7% | 1002 | 15 | 1.5% | 1026 | 14 | 1.4% |
| 43 | 935 | - | - | 959 | 11 | 1.1% | 938 | 14 | 1.5% | 1031 | 15 | 1.5% | 1046 | - | - | 1192 | 22 | 1.8% | 1184 | 13 | 1.1% | 1150 | 15 | 1.3% |
| 44 | 400 | 11 | 2.8% | 521 | - | - | 523 | - | - | 551 | 14 | 2.5% | 723 | 13 | 1.8% | 693 | 15 | 2.2% | 782 | 10 | 1.3% | 755 | 14 | 1.9% |
| 45 | 392 | - | - | 396 | - | - | 413 | - | - | 561 | - | - | 596 | - | - | 627 | - | - | 690 | - | - | 655 | 10 | 1.5% |
| 46 | 519 | 14 | 2.7% | 560 | 14 | 2.5% | 488 | 15 | 3.1% | 637 | 15 | 2.4% | 735 | 20 | 2.7% | 816 | 11 | 1.3% | 903 | 10 | 1.1% | 960 | - | - |
| 47 | 624 | - | - | 630 | - | - | 683 | - | - | 692 | 10 | 1.4% | 712 | - | - | 759 | - | - | 736 | 10 | 1.4% | 743 | - | - |

"-" indicates the value of <10.

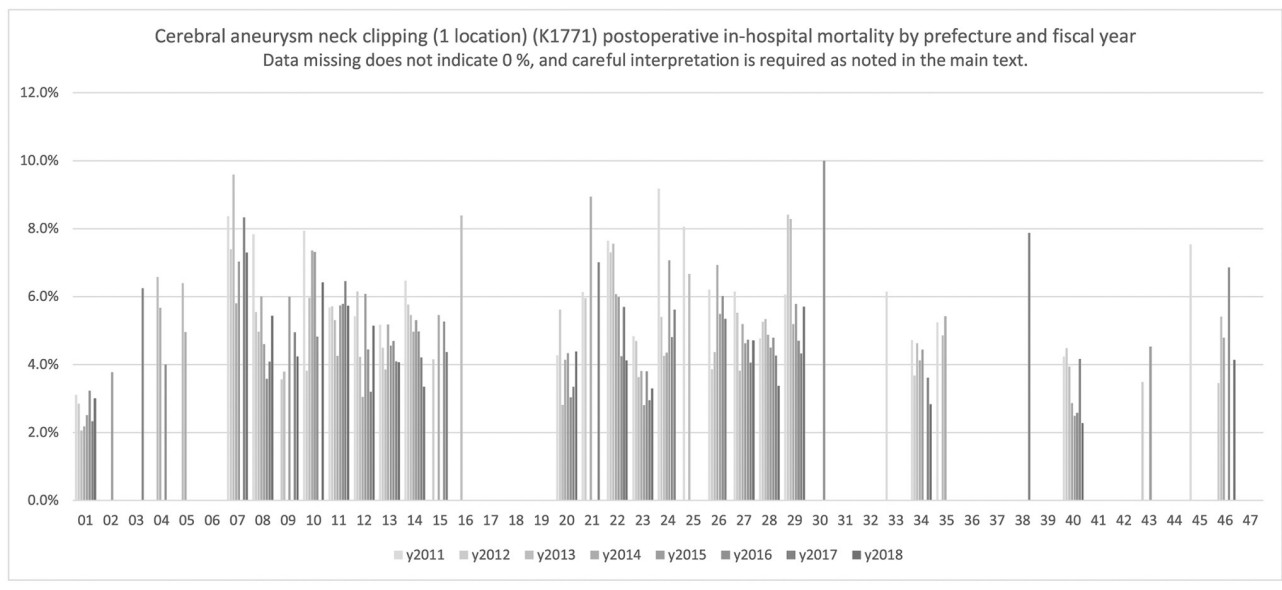

**Fig 2. Cerebral aneurysm neck clipping (one location) (K1771).** Cerebral aneurysm neck clipping (1 location) (K1771) postoperative in-hospital mortality by prefecture and fiscal year.

surgeries; however, the numerical value of outcomes for other coronary artery bypass grafting could not be depicted because the number of each outcome was small.

### 3.4. Tracheotomy (K386)

Fig 4 depicts the postoperative in-hospital mortality, followed by Table 7 shows the results of tracheotomy cases. Data shown in Table 7 included relatively high numerical results for both the denominator and numerator; in several regions, comparisons in terms of temporal changes and regional differences could be made. Although the tabulation was an analysis at the prefectural level, which indicates that the collections include a broad range of data from each region, clear regional differences were depicted in the index values; thus, the implications should be carefully interpreted as presented in the Discussion. The temporal changes do not appear to be uniform among regions, although the results for all regions combined showed a decreasing trend over time.

## 4. Discussion

The present study analyzed surgical outcomes at hospitals throughout Japan according to regional differences and temporal changes using the highly comprehensive DPC database and provided thorough results. Despite the simplicity of the method and broad analysis at the prefectural level, regional differences were observed in all surgical fields. The analysis of the administrative data was useful because this report confirmed the advantage of the Japanese DPC, in which DPC data are accompanied by the creation of Form 1 that corresponded to the hospitalization summary.

The most important aspect in the results of this study was its comprehensiveness in managing data from numerous hospitals. Some reports have indicated that Japan has fewer perioperative complications than other countries [1, 2]; however, from a global perspective, it is of interest to determine whether such trends observed in those reports are the same in hospitals throughout Japan, including hospitals not included in the registry data or whether this is

**Table 2. Cerebral aneurysm neck clipping (1 location; K1771): Case number and outcomes of postoperative in-hospital mortality.**

| Fiscal year | 2011 | | | 2012 | | | 2013 | | | 2014 | | | 2015 | | | 2016 | | | 2017 | | | 2018 | | |
|---|---|---|---|---|---|---|---|---|---|---|---|---|---|---|---|---|---|---|---|---|---|---|---|---|
| Prefecture | Case | Outcome | | Case | Outcome | | Case | Outcome | | Case | Outcome | | Case | Outcome | | Case | Outcome | | Case | Outcome | | Case | Outcome | |
| 01 | 1225 | 38 | 3.1% | 1299 | 37 | 2.8% | 1264 | 26 | 2.1% | 1288 | 28 | 2.2% | 1314 | 33 | 2.5% | 1209 | 39 | 3.2% | 1158 | 27 | 2.3% | 1164 | 35 | 3.0% |
| 02 | 272 | - | - | 258 | - | - | 241 | - | - | 232 | - | - | 265 | 10 | 3.8% | 226 | - | - | 209 | - | - | 171 | - | - |
| 03 | 257 | - | - | 259 | - | - | 241 | - | - | 202 | - | - | 182 | - | - | 176 | - | - | 176 | 11 | 6.3% | 163 | - | - |
| 04 | 197 | - | - | 194 | - | - | 152 | 10 | 6.6% | 194 | 11 | 5.7% | 274 | - | - | 275 | 11 | 4.0% | 264 | - | - | 250 | - | - |
| 05 | 136 | - | - | 117 | - | - | 172 | 11 | 6.4% | 202 | 10 | 5.0% | 178 | - | - | 167 | - | - | 165 | - | - | 145 | - | - |
| 06 | 175 | - | - | 182 | - | - | 156 | - | - | 179 | - | - | 187 | - | - | 157 | - | - | 162 | - | - | 135 | - | - |
| 07 | 263 | 22 | 8.4% | 230 | 17 | 7.4% | 219 | 21 | 9.6% | 207 | 12 | 5.8% | 185 | 13 | 7.0% | 194 | - | - | 204 | 17 | 8.3% | 192 | 14 | 7.3% |
| 08 | 332 | 26 | 7.8% | 433 | 24 | 5.5% | 403 | 20 | 5.0% | 433 | 26 | 6.0% | 456 | 21 | 4.6% | 419 | 15 | 3.6% | 367 | 15 | 4.1% | 405 | 22 | 5.4% |
| 09 | 244 | - | - | 281 | 10 | 3.6% | 290 | 11 | 3.8% | 265 | - | - | 267 | 16 | 6.0% | 250 | - | - | 263 | 13 | 4.9% | 283 | 12 | 4.2% |
| 10 | 252 | 20 | 7.9% | 262 | 10 | 3.8% | 218 | 13 | 6.0% | 272 | 20 | 7.4% | 260 | 19 | 7.3% | 228 | 11 | 4.8% | 210 | - | - | 218 | 14 | 6.4% |
| 11 | 634 | 36 | 5.7% | 683 | 39 | 5.7% | 660 | 35 | 5.3% | 705 | 30 | 4.3% | 645 | 37 | 5.7% | 726 | 42 | 5.8% | 666 | 43 | 6.5% | 698 | 40 | 5.7% |
| 12 | 572 | 31 | 5.4% | 585 | 36 | 6.2% | 592 | 25 | 4.2% | 590 | 18 | 3.1% | 543 | 33 | 6.1% | 540 | 24 | 4.4% | 563 | 18 | 3.2% | 564 | 29 | 5.1% |
| 13 | 1626 | 84 | 5.2% | 1646 | 74 | 4.5% | 1636 | 63 | 3.9% | 1527 | 79 | 5.2% | 1538 | 70 | 4.6% | 1406 | 66 | 4.7% | 1394 | 57 | 4.1% | 1352 | 55 | 4.1% |
| 14 | 1036 | 67 | 6.5% | 989 | 57 | 5.8% | 880 | 48 | 5.5% | 1007 | 50 | 5.0% | 962 | 51 | 5.3% | 986 | 49 | 5.0% | 999 | 42 | 4.2% | 957 | 32 | 3.3% |
| 15 | 219 | - | - | 241 | 10 | 4.1% | 224 | - | - | 220 | 12 | 5.5% | 198 | - | - | 228 | 12 | 5.3% | 229 | 10 | 4.4% | 202 | - | - |
| 16 | 168 | - | - | 131 | - | - | 155 | 13 | 8.4% | 139 | - | - | 127 | - | - | 123 | - | - | 124 | - | - | 137 | - | - |
| 17 | 130 | - | - | 114 | - | - | 112 | - | - | 104 | - | - | 105 | - | - | 104 | - | - | 106 | - | - | 77 | - | - |
| 18 | 94 | - | - | 92 | - | - | 93 | - | - | 91 | - | - | 71 | - | - | 72 | - | - | 59 | - | - | 65 | - | - |
| 19 | 75 | - | - | 76 | - | - | 96 | - | - | 80 | - | - | 83 | - | - | 82 | - | - | 68 | - | - | 74 | - | - |
| 20 | 375 | 16 | 4.3% | 445 | 25 | 5.6% | 427 | 12 | 2.8% | 435 | 18 | 4.1% | 392 | 17 | 4.3% | 362 | 11 | 3.0% | 329 | 11 | 3.3% | 342 | 15 | 4.4% |
| 21 | 212 | 13 | 6.1% | 235 | 14 | 6.0% | 210 | - | - | 179 | 16 | 8.9% | 183 | - | - | 155 | - | - | 157 | 11 | 7.0% | 196 | - | - |
| 22 | 589 | 45 | 7.6% | 548 | 40 | 7.3% | 503 | 38 | 7.6% | 494 | 30 | 6.1% | 501 | 30 | 6.0% | 471 | 20 | 4.2% | 474 | 27 | 5.7% | 437 | 18 | 4.1% |
| 23 | 889 | 43 | 4.8% | 852 | 40 | 4.7% | 854 | 31 | 3.6% | 840 | 32 | 3.8% | 891 | 25 | 2.8% | 816 | 31 | 3.8% | 847 | 25 | 3.0% | 790 | 26 | 3.3% |
| 24 | 218 | 20 | 9.2% | 241 | 13 | 5.4% | 235 | 10 | 4.3% | 230 | 10 | 4.3% | 184 | 13 | 7.1% | 208 | 10 | 4.8% | 196 | 11 | 5.6% | 199 | - | - |
| 25 | 149 | 12 | 8.1% | 127 | - | - | 150 | 10 | 6.7% | 131 | - | - | 130 | - | - | 126 | - | - | 127 | - | - | 124 | - | - |
| 26 | 258 | 16 | 6.2% | 259 | 10 | 3.9% | 252 | 11 | 4.4% | 231 | 16 | 6.9% | 237 | 13 | 5.5% | 216 | 13 | 6.0% | 206 | 11 | 5.3% | 152 | - | - |
| 27 | 1221 | 75 | 6.1% | 1249 | 69 | 5.5% | 1205 | 46 | 3.8% | 1156 | 60 | 5.2% | 1081 | 50 | 4.6% | 1057 | 50 | 4.7% | 1012 | 41 | 4.1% | 870 | 41 | 4.7% |
| 28 | 733 | 35 | 4.8% | 780 | 41 | 5.3% | 825 | 44 | 5.3% | 801 | 39 | 4.9% | 733 | 33 | 4.5% | 751 | 36 | 4.8% | 704 | 30 | 4.3% | 682 | 23 | 3.4% |
| 29 | 198 | 12 | 6.1% | 202 | 17 | 8.4% | 181 | 15 | 8.3% | 212 | 11 | 5.2% | 225 | 13 | 5.8% | 234 | 11 | 4.7% | 231 | 10 | 4.3% | 228 | 13 | 5.7% |
| 30 | 120 | - | - | 98 | - | - | 117 | - | - | 116 | - | - | 96 | - | - | 100 | 10 | 10.0% | 70 | - | - | 76 | - | - |
| 31 | 97 | - | - | 103 | - | - | 81 | - | - | 70 | - | - | 76 | - | - | 52 | - | - | 74 | - | - | 59 | - | - |
| 32 | 93 | - | - | 83 | - | - | 93 | - | - | 83 | - | - | 84 | - | - | 74 | - | - | 75 | - | - | 73 | - | - |
| 33 | 244 | 15 | 6.1% | 242 | - | - | 259 | - | - | 222 | - | - | 208 | - | - | 207 | - | - | 175 | - | - | 171 | - | - |
| 34 | 339 | 16 | 4.7% | 299 | 11 | 3.7% | 324 | 15 | 4.6% | 291 | 12 | 4.1% | 293 | 13 | 4.4% | 291 | - | - | 305 | 11 | 3.6% | 353 | 10 | 2.8% |
| 35 | 191 | 10 | 5.2% | 172 | - | - | 206 | 10 | 4.9% | 203 | 11 | 5.4% | 193 | - | - | 189 | - | - | 153 | - | - | 142 | - | - |
| 36 | 105 | - | - | 93 | - | - | 103 | - | - | 93 | - | - | 70 | - | - | 60 | - | - | 61 | - | - | 73 | - | - |
| 37 | 84 | - | - | 83 | - | - | 84 | - | - | 84 | - | - | 108 | - | - | 74 | - | - | 66 | - | - | 68 | - | - |
| 38 | 135 | - | - | 135 | - | - | 110 | - | - | 177 | - | - | 127 | - | - | 155 | - | - | 165 | 13 | 7.9% | 125 | - | - |
| 39 | 120 | - | - | 96 | - | - | 120 | - | - | 82 | - | - | 83 | - | - | 96 | - | - | 104 | - | - | 94 | - | - |
| 40 | 803 | 34 | 4.2% | 803 | 36 | 4.5% | 711 | 28 | 3.9% | 768 | 22 | 2.9% | 723 | 18 | 2.5% | 621 | 16 | 2.6% | 624 | 26 | 4.2% | 616 | 14 | 2.3% |
| 41 | 102 | - | - | 99 | - | - | 104 | - | - | 136 | - | - | 118 | - | - | 118 | - | - | 115 | - | - | 102 | - | - |
| 42 | 229 | - | - | 207 | - | - | 192 | - | - | 178 | - | - | 185 | - | - | 198 | - | - | 195 | - | - | 202 | - | - |
| 43 | 343 | - | - | 316 | 11 | 3.5% | 306 | - | - | 330 | - | - | 287 | 13 | 4.5% | 265 | - | - | 293 | - | - | 242 | - | - |
| 44 | 106 | - | - | 115 | - | - | 89 | - | - | 73 | - | - | 101 | - | - | 118 | - | - | 90 | - | - | 85 | - | - |
| 45 | 146 | 11 | 7.5% | 168 | - | - | 145 | - | - | 108 | - | - | 151 | - | - | 170 | - | - | 151 | - | - | 143 | - | - |
| 46 | 301 | - | - | 290 | 10 | 3.4% | 296 | 16 | 5.4% | 292 | 14 | 4.8% | 290 | - | - | 248 | 17 | 6.9% | 242 | - | - | 242 | 10 | 4.1% |
| 47 | 144 | - | - | 182 | - | - | 124 | - | - | 121 | - | - | 127 | - | - | 130 | - | - | 151 | - | - | 134 | - | - |

"-" indicates the value of <10.

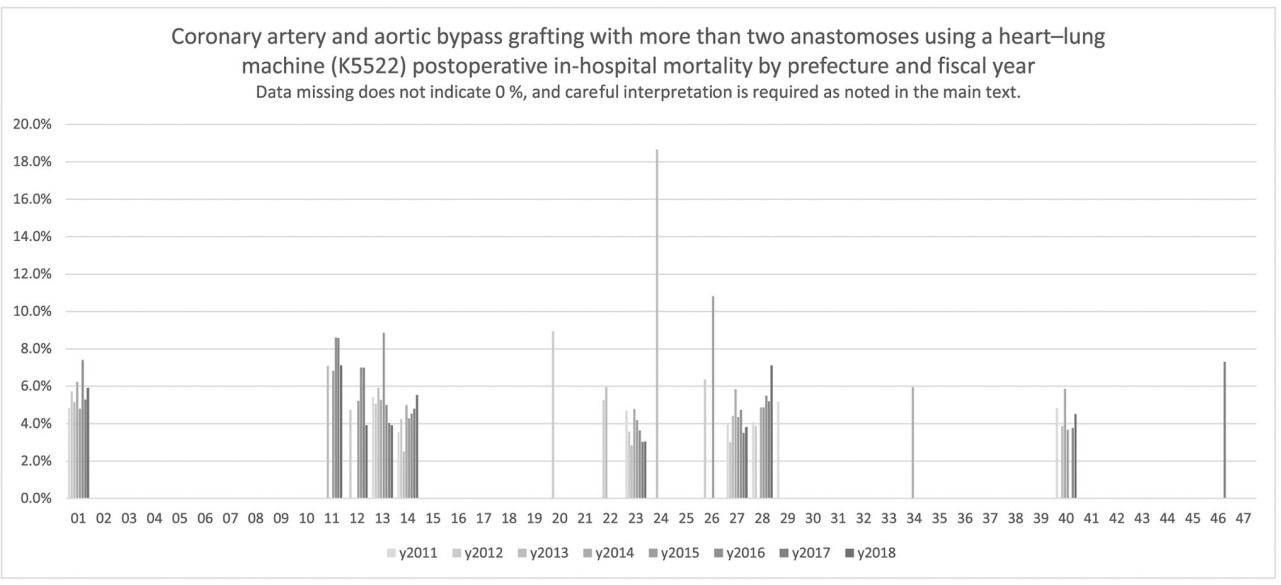

**Fig 3. Coronary artery and aortic bypass grafting with more than two anastomoses using a heart–lung machine (K5522).** Coronary artery and aortic bypass grafting with more than two anastomoses using a heart–lung machine (K5522) postoperative in-hospital mortality by prefecture and fiscal year.

common trends across all surgical fields. Data available from the DPC database, which now accumulates data from numerous Japanese hospitals, facilitate the extensive analysis of actual conditions that may not always be collectible by a registry or other sources. The database enables the investigation of actual implementation of procedures in a manner that was not possible before. In addition, all surgeries were included in this analysis, although the interpretation of the results varied and caution should be exercised while inferring them. Complete results are presented as S1 Data.

In the current study, the results from only some selections from all fields were presented; however, in the complete results, regional differences and temporal changes could be easily observed to some extent. Moreover, negative results about records without noticeable differences or those that remained unchanged was important for review. Although it depends on the purpose of the future research or project, the availability of data in all fields, such as that presented in this study, is considered extremely valuable.

The present study has several limitations and important caveats. In terms of data, several missing values were noted. For example, if the number of cases was 300, 3% of the outcomes was only 9 cases; this value was not obtainable as an aggregate value. In this study the data was compared at a general level, i.e., the prefectural level for analyzing the data of each year, as the author considered it reasonable to obtain tangible results; however, the number of nulls encountered were more than expected. A wider geographic aggregation could have been used to resolve the issue; however, this approach was not used in this study as the author, instead, preferred to conduct comparisons at a more detailed smaller regional level than the prefecture level or at a specific time span. Another issue encountered during the analysis was that the surgical code for artificial head insertion (K0811) included the surgery for both the shoulder and hip. A separate analysis should be conducted by combining disease names or using different master codes from different DPC datasets. However, a simple detailed analysis will yield fewer figures that can be obtained or presented. Although the present study conducted a relatively coarse unit-level analysis, the results showed some trends and differences and can be useful for

**Table 3. Coronary artery and aortic bypass grafting with one anastomosis using a heart–lung machine (K5521): Case number and outcomes of postoperative in-hospital mortality.**

| Fiscal year | 2011 | | | 2012 | | | 2013 | | | 2014 | | | 2015 | | | 2016 | | | 2017 | | | 2018 | | |
|---|---|---|---|---|---|---|---|---|---|---|---|---|---|---|---|---|---|---|---|---|---|---|---|---|
| Prefecture | Case | Outcome | | Case | Outcome | | Case | Outcome | | Case | Outcome | | Case | Outcome | | Case | Outcome | | Case | Outcome | | Case | Outcome | |
| 01 | 20 | - | - | 18 | - | - | 11 | - | - | 17 | - | - | 23 | - | - | 13 | - | - | 15 | - | - | 15 | - | - |
| 02 | - | - | - |  |  |  |  |  |  |  |  |  | - | - | - | - | - | - | - | - | - |  |  |  |
| 03 |  |  |  |  |  |  | - | - | - |  |  |  | - | - | - | - | - | - | - | - | - | - | - | - |
| 04 | - | - | - | - | - | - | 13 | - | - | 11 | - | - | - | - | - | - | - | - | - | - | - | 16 | - | - |
| 05 | - | - | - |  |  |  |  |  |  |  |  |  | - | - | - | - | - | - | - | - | - | - | - | - |
| 06 | - | - | - |  |  |  | - | - | - | - | - | - | - | - | - | - | - | - |  |  |  | - | - | - |
| 07 | - | - | - | - | - | - | - | - | - |  |  |  | - | - | - | - | - | - | - | - | - | - | - | - |
| 08 | - | - | - | - | - | - | - | - | - | - | - | - | - | - | - | - | - | - | - | - | - | - | - | - |
| 09 | - | - | - | - | - | - | - | - | - | - | - | - | - | - | - | - | - | - | - | - | - | - | - | - |
| 10 | - | - | - | - | - | - | - | - | - | - | - | - | - | - | - | - | - | - | - | - | - | - | - | - |
| 11 | 19 | - | - | - | - | - | 15 | - | - | 16 | - | - | - | - | - | 18 | - | - | 17 | - | - | 16 | - | - |
| 12 | - | - | - | 16 | - | - | - | - | - | 16 | - | - | 18 | - | - | 18 | - | - | 16 | - | - | 16 | - | - |
| 13 | 37 | - | - | - | - | - | 44 | - | - | 44 | - | - | 27 | - | - | 41 | - | - | 44 | - | - | 43 | - | - |
| 14 | 27 | - | - | 35 | - | - | 23 | - | - | 24 | - | - | 52 | - | - | 38 | - | - | 34 | - | - | 33 | - | - |
| 15 | - | - | - | 21 | - | - | - | - | - | - | - | - | - | - | - |  |  |  | - | - | - | - | - | - |
| 16 | - | - | - | - | - | - | - | - | - | - | - | - | - | - | - | - | - | - | - | - | - | - | - | - |
| 17 | - | - | - | - | - | - | - | - | - | - | - | - | - | - | - | - | - | - | - | - | - | - | - | - |
| 18 | - | - | - | - | - | - | - | - | - | - | - | - | - | - | - | - | - | - | - | - | - | - | - | - |
| 19 | - | - | - | - | - | - | - | - | - | - | - | - | - | - | - | - | - | - | - | - | - | - | - | - |
| 20 | - | - | - | - | - | - | - | - | - | - | - | - | - | - | - | - | - | - | - | - | - | - | - | - |
| 21 | - | - | - | - | - | - | - | - | - | 10 | - | - | - | - | - | - | - | - | - | - | - | - | - | - |
| 22 | 12 | - | - | - | - | - | 10 | - | - | - | - | - | 12 | - | - | 10 | - | - | - | - | - | - | - | - |
| 23 | 13 | - | - | 14 | - | - | 19 | - | - | 16 | - | - | 24 | - | - | 27 | - | - | 25 | - | - | 35 | - | - |
| 24 | - | - | - | - | - | - | - | - | - | - | - | - | - | - | - | - | - | - | - | - | - | - | - | - |
| 25 | - | - | - | - | - | - | - | - | - | - | - | - | - | - | - | - | - | - | - | - | - | - | - | - |
| 26 | 10 | - | - | - | - | - | - | - | - | - | - | - | - | - | - | - | - | - | - | - | - | - | - | - |
| 27 | 28 | - | - | 24 | - | - | 20 | - | - | 19 | - | - | 33 | - | - | 23 | - | - | 35 | - | - | 25 | - | - |
| 28 | 13 | - | - | 12 | - | - | - | - | - | 11 | - | - | 15 | - | - | 12 | - | - | - | - | - | 14 | - | - |
| 29 | 12 | - | - | - | - | - | - | - | - | - | - | - | - | - | - | - | - | - | - | - | - | - | - | - |
| 30 | - | - | - | - | - | - | - | - | - | - | - | - | - | - | - |  |  |  | - | - | - | - | - | - |
| 31 |  |  |  | - | - | - | - | - | - | - | - | - | - | - | - | - | - | - | - | - | - |  |  |  |
| 32 | - | - | - |  |  |  | - | - | - | - | - | - | - | - | - | - | - | - | - | - | - |  |  |  |
| 33 | - | - | - |  |  |  | - | - | - | - | - | - |  |  |  | - | - | - | - | - | - | - | - | - |
| 34 | - | - | - | - | - | - | 14 | - | - | - | - | - | 10 | - | - | 11 | - | - | 12 | - | - | - | - | - |
| 35 | - | - | - | - | - | - | - | - | - | - | - | - | - | - | - | - | - | - | - | - | - | - | - | - |
| 36 | - | - | - |  |  |  |  |  |  |  |  |  |  |  |  |  |  |  |  |  |  | - | - | - |
| 37 | - | - | - | - | - | - | - | - | - | - | - | - |  |  |  | - | - | - | - | - | - | - | - | - |
| 38 | - | - | - | - | - | - | - | - | - | - | - | - | - | - | - | - | - | - | - | - | - | - | - | - |
| 39 | - | - | - | - | - | - | - | - | - | - | - | - | - | - | - | - | - | - | - | - | - |  |  |  |
| 40 | 15 | - | - | 17 | - | - | - | - | - | 12 | - | - | 13 | - | - | 30 | - | - | 25 | - | - | 14 | - | - |
| 41 | - | - | - | - | - | - | - | - | - | - | - | - | - | - | - | - | - | - | - | - | - | - | - | - |
| 42 | - | - | - | - | - | - | - | - | - | - | - | - | - | - | - | - | - | - | - | - | - | - | - | - |
| 43 | 11 | - | - |  |  |  | - | - | - | - | - | - | - | - | - | - | - | - | - | - | - |  |  |  |
| 44 |  |  |  |  |  |  | - | - | - | - | - | - | - | - | - | - | - | - | - | - | - | - | - | - |
| 45 | - | - | - | - | - | - | - | - | - | - | - | - | - | - | - | - | - | - | - | - | - | - | - | - |
| 46 | - | - | - | - | - | - | - | - | - | - | - | - | - | - | - | - | - | - | - | - | - | - | - | - |
| 47 | - | - | - | - | - | - | - | - | - | - | - | - | 10 | - | - | - | - | - | 11 | - | - | - | - | - |

"-" indicates the value of <10.

**Table 4. Coronary artery and aortic bypass grafting with >two anastomoses using a heart–lung machine (K5522): Case number and outcomes of postoperative in-hospital mortality.**

| Fiscal year | 2011 | | | 2012 | | | 2013 | | | 2014 | | | 2015 | | | 2016 | | | 2017 | | | 2018 | | |
|---|---|---|---|---|---|---|---|---|---|---|---|---|---|---|---|---|---|---|---|---|---|---|---|---|
| Prefecture | Case | Outcome | | Case | Outcome | | Case | Outcome | | Case | Outcome | | Case | Outcome | | Case | Outcome | | Case | Outcome | | Case | Outcome | |
| 01 | 248 | 12 | 4.8% | 280 | 16 | 5.7% | 271 | 14 | 5.2% | 354 | 22 | 6.2% | 313 | 15 | 4.8% | 338 | 25 | 7.4% | 322 | 17 | 5.3% | 338 | 20 | 5.9% |
| 02 | 31 | - | - | 37 | - | - | 35 | - | - | 34 | - | - | 44 | - | - | 37 | - | - | 49 | - | - | 39 | - | - |
| 03 | 20 | - | - | 20 | - | - | 26 | - | - | 23 | - | - | 15 | - | - | 20 | - | - | 25 | - | - | 32 | - | - |
| 04 | 144 | - | - | 165 | - | - | 142 | - | - | 159 | - | - | 155 | - | - | 175 | - | - | 187 | - | - | 167 | - | - |
| 05 | 19 | - | - | 20 | - | - | 26 | - | - | 32 | - | - | 33 | - | - | 36 | - | - | 31 | - | - | 30 | - | - |
| 06 | 54 | - | - | 60 | - | - | 66 | - | - | 46 | - | - | 50 | - | - | 51 | - | - | 61 | - | - | 71 | - | - |
| 07 | 48 | - | - | 46 | - | - | 63 | - | - | 62 | - | - | 59 | - | - | 68 | - | - | 73 | - | - | 56 | - | - |
| 08 | 86 | - | - | 87 | - | - | 84 | - | - | 78 | - | - | 68 | - | - | 95 | - | - | 118 | - | - | 97 | - | - |
| 09 | 149 | - | - | 130 | - | - | 155 | - | - | 134 | - | - | 165 | - | - | 156 | - | - | 147 | - | - | 120 | - | - |
| 10 | 137 | - | - | 145 | - | - | 109 | - | - | 110 | - | - | 93 | - | - | 96 | - | - | 104 | - | - | 133 | - | - |
| 11 | 123 | - | - | 151 | - | - | 141 | 10 | 7.1% | 161 | - | - | 176 | 12 | 6.8% | 186 | 16 | 8.6% | 198 | 17 | 8.6% | 239 | 17 | 7.1% |
| 12 | 250 | - | - | 316 | 15 | 4.7% | 290 | - | - | 244 | - | - | 249 | 13 | 5.2% | 272 | 19 | 7.0% | 272 | 19 | 7.0% | 256 | 10 | 3.9% |
| 13 | 720 | 39 | 5.4% | 731 | 37 | 5.1% | 676 | 40 | 5.9% | 702 | 37 | 5.3% | 644 | 57 | 8.9% | 720 | 36 | 5.0% | 744 | 30 | 4.0% | 665 | 26 | 3.9% |
| 14 | 424 | 15 | 3.5% | 471 | 20 | 4.2% | 480 | 12 | 2.5% | 420 | 21 | 5.0% | 444 | 19 | 4.3% | 375 | 17 | 4.5% | 417 | 20 | 4.8% | 380 | 21 | 5.5% |
| 15 | 83 | - | - | 103 | - | - | 81 | - | - | 76 | - | - | 63 | - | - | 49 | - | - | 43 | - | - | 55 | - | - |
| 16 | 49 | - | - | 48 | - | - | 44 | - | - | 38 | - | - | 45 | - | - | 56 | - | - | 50 | - | - | 36 | - | - |
| 17 | 63 | - | - | 64 | - | - | 53 | - | - | 70 | - | - | 75 | - | - | 66 | - | - | 51 | - | - | 51 | - | - |
| 18 | 83 | - | - | 71 | - | - | 74 | - | - | 67 | - | - | 58 | - | - | 76 | - | - | 62 | - | - | 42 | - | - |
| 19 | 77 | - | - | 60 | - | - | 63 | - | - | 60 | - | - | 67 | - | - | 69 | - | - | 68 | - | - | 77 | - | - |
| 20 | 114 | - | - | 123 | 11 | 8.9% | 114 | - | - | 113 | - | - | 136 | - | - | 118 | - | - | 111 | - | - | 103 | - | - |
| 21 | 36 | - | - | 52 | - | - | 59 | - | - | 80 | - | - | 69 | - | - | 79 | - | - | 90 | - | - | 96 | - | - |
| 22 | 152 | - | - | 190 | 10 | 5.3% | 185 | 11 | 5.9% | 168 | - | - | 203 | - | - | 173 | - | - | 171 | - | - | 169 | - | - |
| 23 | 405 | 19 | 4.7% | 449 | 16 | 3.6% | 390 | 11 | 2.8% | 501 | 24 | 4.8% | 502 | 21 | 4.2% | 578 | 21 | 3.6% | 564 | 17 | 3.0% | 561 | 17 | 3.0% |
| 24 | 60 | - | - | 74 | - | - | 59 | 11 | 18.6% | 35 | - | - | 55 | - | - | 51 | - | - | 89 | - | - | 78 | - | - |
| 25 | 30 | - | - | 47 | - | - | 34 | - | - | 35 | - | - | 31 | - | - | 40 | - | - | 47 | - | - | 29 | - | - |
| 26 | 138 | - | - | 173 | 11 | 6.4% | 175 | - | - | 137 | - | - | 111 | 12 | 10.8% | 126 | - | - | 125 | - | - | 108 | - | - |
| 27 | 584 | 23 | 3.9% | 632 | 19 | 3.0% | 590 | 26 | 4.4% | 566 | 33 | 5.8% | 644 | 28 | 4.3% | 654 | 31 | 4.7% | 628 | 22 | 3.5% | 683 | 26 | 3.8% |
| 28 | 319 | 13 | 4.1% | 335 | 13 | 3.9% | 341 | - | - | 370 | 18 | 4.9% | 287 | 14 | 4.9% | 291 | 16 | 5.5% | 289 | 15 | 5.2% | 267 | 19 | 7.1% |
| 29 | 193 | 10 | 5.2% | 199 | - | - | 176 | - | - | 155 | - | - | 130 | - | - | 120 | - | - | 122 | - | - | 118 | - | - |
| 30 | 97 | - | - | 97 | - | - | 88 | - | - | 113 | - | - | 75 | - | - | 86 | - | - | 69 | - | - | 68 | - | - |
| 31 | 51 | - | - | 55 | - | - | 59 | - | - | 45 | - | - | 57 | - | - | 58 | - | - | 60 | - | - | 66 | - | - |
| 32 | 28 | - | - | 28 | - | - | 31 | - | - | 38 | - | - | 50 | - | - | 54 | - | - | 41 | - | - | 34 | - | - |
| 33 | 133 | - | - | 158 | - | - | 192 | - | - | 148 | - | - | 146 | - | - | 150 | - | - | 148 | - | - | 128 | - | - |
| 34 | 84 | - | - | 94 | - | - | 91 | - | - | 168 | 10 | 6.0% | 203 | - | - | 142 | - | - | 131 | - | - | 130 | - | - |
| 35 | 65 | - | - | 63 | - | - | 43 | - | - | 55 | - | - | 49 | - | - | 76 | - | - | 75 | - | - | 60 | - | - |
| 36 | 39 | - | - | 27 | - | - | 21 | - | - | 18 | - | - | 20 | - | - | 26 | - | - | 24 | - | - | 22 | - | - |
| 37 | 23 | - | - | 41 | - | - | 45 | - | - | 56 | - | - | 55 | - | - | 48 | - | - | 34 | - | - | 41 | - | - |
| 38 | 28 | - | - | 28 | - | - | 30 | - | - | 54 | - | - | 37 | - | - | 67 | - | - | 71 | - | - | 51 | - | - |
| 39 | 76 | - | - | 57 | - | - | 80 | - | - | 84 | - | - | 95 | - | - | 101 | - | - | 86 | - | - | 72 | - | - |
| 40 | 269 | 13 | 4.8% | 254 | - | - | 259 | 10 | 3.9% | 307 | 18 | 5.9% | 300 | 11 | 3.7% | 307 | - | - | 398 | 15 | 3.8% | 355 | 16 | 4.5% |
| 41 | 41 | - | - | 36 | - | - | 29 | - | - | 31 | - | - | 36 | - | - | 52 | - | - | 49 | - | - | 42 | - | - |
| 42 | 161 | - | - | 147 | - | - | 168 | - | - | 166 | - | - | 136 | - | - | 115 | - | - | 139 | - | - | 136 | - | - |
| 43 | 123 | - | - | 143 | - | - | 153 | - | - | 140 | - | - | 159 | - | - | 128 | - | - | 97 | - | - | 97 | - | - |
| 44 | 24 | - | - | 28 | - | - | 21 | - | - | 22 | - | - | 32 | - | - | 32 | - | - | 32 | - | - | 30 | - | - |
| 45 | 84 | - | - | 83 | - | - | 88 | - | - | 100 | - | - | 93 | - | - | 79 | - | - | 76 | - | - | 94 | - | - |
| 46 | 109 | - | - | 154 | - | - | 129 | - | - | 115 | - | - | 144 | - | - | 137 | - | - | 137 | 10 | 7.3% | 68 | - | - |
| 47 | 59 | - | - | 58 | - | - | 53 | - | - | 84 | - | - | 92 | - | - | 87 | - | - | 91 | - | - | 99 | - | - |

"-" indicates the value of <10.

**Table 5. Coronary artery and aortic bypass grafting with one anastomosis without a heart–lung machine (K552-21): Case number and outcomes of postoperative in-hospital mortality.**

| Fiscal year | 2011 | | | 2012 | | | 2013 | | | 2014 | | | 2015 | | | 2016 | | | 2017 | | | 2018 | | |
|---|---|---|---|---|---|---|---|---|---|---|---|---|---|---|---|---|---|---|---|---|---|---|---|---|
| Prefecture | Case | Outcome | | Case | Outcome | | Case | Outcome | | Case | Outcome | | Case | Outcome | | Case | Outcome | | Case | Outcome | | Case | Outcome | |
| 01 | 46 | - | - | 60 | - | - | 60 | - | - | 51 | - | - | 42 | - | - | 84 | - | - | 83 | - | - | 52 | - | - |
| 02 | 15 | - | - | 13 | - | - | 21 | - | - | 25 | - | - | 11 | - | - | 15 | - | - | 13 | - | - | - | - | - |
| 03 | 10 | - | - | 15 | - | - | - | - | - | - | - | - | 13 | - | - | - | - | - | 16 | - | - | 14 | - | - |
| 04 | 13 | - | - | 11 | - | - | 13 | - | - | 11 | - | - | 11 | - | - | 14 | - | - | 11 | - | - | 18 | - | - |
| 05 | 14 | - | - | - | - | - | - | - | - | - | - | - | - | - | - | - | - | - | - | - | - | - | - | - |
| 06 | 20 | - | - | 14 | - | - | - | - | - | 12 | - | - | 12 | - | - | - | - | - | - | - | - | 12 | - | - |
| 07 | - | - | - | 13 | - | - | - | - | - | - | - | - | 10 | - | - | - | - | - | - | - | - | - | - | - |
| 08 | - | - | - | - | - | - | 15 | - | - | 13 | - | - | - | - | - | 15 | - | - | 12 | - | - | - | - | - |
| 09 | - | - | - | - | - | - | - | - | - | - | - | - | - | - | - | - | - | - | - | - | - | - | - | - |
| 10 | 19 | - | - | 18 | - | - | 14 | - | - | 13 | - | - | 10 | - | - | 15 | - | - | - | - | - | - | - | - |
| 11 | 20 | - | - | 19 | - | - | 19 | - | - | 26 | - | - | 31 | - | - | 22 | - | - | 23 | - | - | 29 | - | - |
| 12 | 33 | - | - | 42 | - | - | 24 | - | - | 25 | - | - | 34 | - | - | 36 | - | - | 60 | - | - | 79 | - | - |
| 13 | 68 | - | - | 100 | - | - | 90 | - | - | 104 | - | - | 102 | - | - | 73 | - | - | 95 | - | - | 107 | - | - |
| 14 | 59 | - | - | 52 | - | - | 51 | - | - | 59 | - | - | 52 | - | - | 66 | - | - | 73 | - | - | 78 | - | - |
| 15 | - | - | - | - | - | - | - | - | - | - | - | - | - | - | - | - | - | - | - | - | - | - | - | - |
| 16 | - | - | - | - | - | - | - | - | - | - | - | - | - | - | - | 10 | - | - | - | - | - | - | - | - |
| 17 | - | - | - | 10 | - | - | - | - | - | - | - | - | - | - | - | - | - | - | - | - | - | - | - | - |
| 18 | - | - | - | - | - | - | - | - | - | - | - | - | - | - | - | - | - | - | - | - | - | - | - | - |
| 19 | - | - | - | - | - | - | | | | - | - | - | - | - | - | - | - | - | | | | - | - | - |
| 20 | 20 | - | - | 29 | - | - | 25 | - | - | 24 | - | - | 19 | - | - | 17 | - | - | 25 | - | - | 24 | - | - |
| 21 | 17 | - | - | 10 | - | - | 14 | - | - | - | - | - | 14 | - | - | 16 | - | - | - | - | - | 20 | - | - |
| 22 | 11 | - | - | 13 | - | - | 30 | - | - | 29 | - | - | 32 | - | - | 30 | - | - | 26 | - | - | 22 | - | - |
| 23 | 66 | - | - | 60 | - | - | 76 | - | - | 54 | - | - | 72 | - | - | 70 | - | - | 59 | - | - | 46 | - | - |
| 24 | - | - | - | - | - | - | - | - | - | - | - | - | 14 | - | - | - | - | - | 11 | - | - | 15 | - | - |
| 25 | - | - | - | - | - | - | - | - | - | - | - | - | 12 | - | - | - | - | - | - | - | - | - | - | - |
| 26 | 12 | - | - | 18 | - | - | 13 | - | - | 10 | - | - | - | - | - | 13 | - | - | 14 | - | - | - | - | - |
| 27 | 38 | - | - | 66 | - | - | 42 | - | - | 39 | - | - | 46 | - | - | 36 | - | - | 42 | - | - | 40 | - | - |
| 28 | 15 | - | - | 21 | - | - | 20 | - | - | 34 | - | - | 19 | - | - | 23 | - | - | 18 | - | - | 31 | - | - |
| 29 | - | - | - | - | - | - | - | - | - | - | - | - | - | - | - | - | - | - | - | - | - | - | - | - |
| 30 | - | - | - | - | - | - | - | - | - | - | - | - | - | - | - | 12 | - | - | - | - | - | - | - | - |
| 31 | - | - | - | - | - | - | - | - | - | - | - | - | - | - | - | - | - | - | - | - | - | - | - | - |
| 32 | - | - | - | - | - | - | - | - | - | - | - | - | - | - | - | - | - | - | - | - | - | 10 | - | - |
| 33 | 13 | - | - | 18 | - | - | 13 | - | - | 19 | - | - | 21 | - | - | 22 | - | - | 11 | - | - | 14 | - | - |
| 34 | 18 | - | - | 20 | - | - | 18 | - | - | 14 | - | - | 15 | - | - | 18 | - | - | 16 | - | - | 16 | - | - |
| 35 | - | - | - | 18 | - | - | 14 | - | - | 20 | - | - | 10 | - | - | 18 | - | - | - | - | - | - | - | - |
| 36 | - | - | - | - | - | - | - | - | - | | | | - | - | - | - | - | - | - | - | - | - | - | - |
| 37 | - | - | - | 13 | - | - | - | - | - | 11 | - | - | - | - | - | - | - | - | - | - | - | - | - | - |
| 38 | - | - | - | 10 | - | - | - | - | - | 11 | - | - | 14 | - | - | - | - | - | - | - | - | 11 | - | - |
| 39 | - | - | - | | | | - | - | - | - | - | - | - | - | - | - | - | - | - | - | - | - | - | - |
| 40 | 37 | - | - | 21 | - | - | 20 | - | - | 19 | - | - | 30 | - | - | 18 | - | - | 18 | - | - | 14 | - | - |
| 41 | - | - | - | - | - | - | - | - | - | - | - | - | - | - | - | - | - | - | - | - | - | - | - | - |
| 42 | 12 | - | - | 11 | - | - | - | - | - | - | - | - | - | - | - | - | - | - | - | - | - | - | - | - |
| 43 | - | - | - | 13 | - | - | - | - | - | 11 | - | - | - | - | - | - | - | - | - | - | - | - | - | - |
| 44 | - | - | - | - | - | - | - | - | - | - | - | - | - | - | - | - | - | - | 14 | - | - | 19 | - | - |
| 45 | | | | - | - | - | - | - | - | - | - | - | - | - | - | - | - | - | - | - | - | - | - | - |
| 46 | - | - | - | - | - | - | - | - | - | - | - | - | 15 | - | - | - | - | - | - | - | - | - | - | - |
| 47 | - | - | - | 12 | - | - | 16 | - | - | 19 | - | - | 13 | - | - | 11 | - | - | 12 | - | - | - | - | - |

"-" indicates the value of <10.

**Table 6. Coronary artery and aortic bypass grafting with >two anastomoses without a heart–lung machine (K552-22): Case number and outcomes of postoperative in-hospital mortality.**

| Fiscal year | 2011 | | | 2012 | | | 2013 | | | 2014 | | | 2015 | | | 2016 | | | 2017 | | | 2018 | | |
|---|---|---|---|---|---|---|---|---|---|---|---|---|---|---|---|---|---|---|---|---|---|---|---|---|
| Prefecture | Case | Outcome | | Case | Outcome | | Case | Outcome | | Case | Outcome | | Case | Outcome | | Case | Outcome | | Case | Outcome | | Case | Outcome | |
| 01 | 414 | 10 | 2.4% | 477 | - | - | 451 | 13 | 2.9% | 469 | - | - | 562 | - | - | 528 | - | - | 545 | - | - | 471 | 13 | 2.8% |
| 02 | 135 | - | - | 109 | - | - | 115 | - | - | 82 | - | - | 121 | - | - | 89 | - | - | 94 | - | - | 107 | - | - |
| 03 | 95 | - | - | 80 | - | - | 68 | - | - | 80 | - | - | 54 | - | - | 73 | - | - | 85 | - | - | 99 | - | - |
| 04 | 49 | - | - | 62 | - | - | 38 | - | - | 31 | - | - | 57 | - | - | 57 | - | - | 45 | - | - | 42 | - | - |
| 05 | 29 | - | - | 29 | - | - | 19 | - | - | 15 | - | - | 16 | - | - | - | - | - | 10 | - | - | 11 | - | - |
| 06 | 28 | - | - | 38 | - | - | 45 | - | - | 55 | - | - | 68 | - | - | 37 | - | - | 42 | - | - | 58 | - | - |
| 07 | 71 | - | - | 76 | - | - | 68 | - | - | 75 | - | - | 80 | - | - | 88 | - | - | 53 | - | - | 67 | - | - |
| 08 | 103 | - | - | 94 | - | - | 103 | - | - | 94 | - | - | 105 | - | - | 115 | - | - | 96 | - | - | 121 | - | - |
| 09 | 20 | - | - | 18 | - | - | 34 | - | - | 26 | - | - | 12 | - | - | 20 | - | - | 24 | - | - | 19 | - | - |
| 10 | 52 | - | - | 78 | - | - | 97 | - | - | 66 | - | - | 72 | - | - | 43 | - | - | 34 | - | - | 33 | - | - |
| 11 | 361 | - | - | 354 | - | - | 351 | - | - | 312 | - | - | 343 | - | - | 364 | - | - | 350 | - | - | 361 | - | - |
| 12 | 183 | - | - | 215 | - | - | 272 | - | - | 359 | - | - | 363 | - | - | 379 | - | - | 345 | - | - | 366 | - | - |
| 13 | 1024 | 14 | 1.4% | 1178 | 10 | 0.8% | 1148 | 21 | 1.8% | 1168 | 13 | 1.1% | 1144 | 15 | 1.3% | 1048 | 14 | 1.3% | 1083 | 15 | 1.4% | 1083 | - | - |
| 14 | 339 | - | - | 331 | - | - | 318 | - | - | 391 | - | - | 475 | - | - | 503 | 10 | 2.0% | 425 | - | - | 424 | - | - |
| 15 | 84 | - | - | 99 | - | - | 82 | - | - | 115 | - | - | 90 | - | - | 67 | - | - | 70 | - | - | 49 | - | - |
| 16 | 88 | - | - | 79 | - | - | 94 | - | - | 102 | - | - | 110 | - | - | 83 | - | - | 63 | - | - | 92 | - | - |
| 17 | 84 | - | - | 85 | - | - | 75 | - | - | 44 | - | - | 70 | - | - | 89 | - | - | 100 | - | - | 98 | - | - |
| 18 | 29 | - | - | 38 | - | - | 46 | - | - | 28 | - | - | 48 | - | - | 30 | - | - | | | | 38 | - | - |
| 19 | | | | | | | | | | - | - | - | - | - | - | - | - | - | 37 | - | - | - | - | - |
| 20 | 122 | - | - | 129 | - | - | 137 | - | - | 132 | - | - | 134 | - | - | 123 | - | - | 99 | - | - | 113 | - | - |
| 21 | 40 | - | - | 70 | - | - | 42 | - | - | 43 | - | - | 86 | - | - | 117 | - | - | 119 | - | - | 137 | - | - |
| 22 | 218 | - | - | 256 | - | - | 236 | - | - | 235 | - | - | 219 | 10 | 4.6% | 223 | - | - | 218 | - | - | 226 | - | - |
| 23 | 466 | - | - | 512 | - | - | 547 | - | - | 496 | - | - | 490 | - | - | 495 | - | - | 502 | - | - | 518 | 11 | 2.1% |
| 24 | 151 | - | - | 166 | - | - | 154 | - | - | 166 | - | - | 143 | - | - | 127 | - | - | 116 | - | - | 126 | - | - |
| 25 | 139 | - | - | 136 | - | - | 130 | - | - | 159 | - | - | 131 | - | - | 125 | - | - | 129 | - | - | 116 | - | - |
| 26 | 154 | - | - | 156 | - | - | 129 | - | - | 120 | - | - | 144 | - | - | 127 | - | - | 133 | - | - | 140 | - | - |
| 27 | 443 | - | - | 492 | - | - | 507 | 10 | 2.0% | 453 | 10 | 2.2% | 478 | - | - | 425 | - | - | 421 | - | - | 381 | - | - |
| 28 | 179 | - | - | 233 | - | - | 255 | - | - | 250 | - | - | 273 | - | - | 202 | - | - | 187 | - | - | 236 | - | - |
| 29 | 34 | - | - | 17 | - | - | 40 | - | - | 48 | - | - | 50 | - | - | 61 | - | - | 57 | - | - | 60 | - | - |
| 30 | 74 | - | - | 98 | - | - | 134 | - | - | 110 | - | - | 97 | - | - | 95 | - | - | 102 | - | - | 103 | - | - |
| 31 | 21 | - | - | 39 | - | - | 17 | - | - | 24 | - | - | 21 | - | - | 16 | - | - | 16 | - | - | 12 | - | - |
| 32 | 23 | - | - | 28 | - | - | 31 | - | - | 34 | - | - | 32 | - | - | 22 | - | - | 19 | - | - | 28 | - | - |
| 33 | 145 | - | - | 170 | - | - | 182 | - | - | 184 | - | - | 128 | - | - | 122 | - | - | 132 | - | - | 124 | - | - |
| 34 | 168 | - | - | 148 | - | - | 135 | - | - | 103 | - | - | 63 | - | - | 111 | - | - | 83 | - | - | 76 | - | - |
| 35 | 102 | - | - | 107 | - | - | 120 | - | - | 148 | - | - | 98 | - | - | 110 | - | - | 124 | - | - | 84 | - | - |
| 36 | 88 | - | - | 80 | - | - | 64 | - | - | 64 | - | - | 62 | - | - | 58 | - | - | 54 | - | - | 50 | - | - |
| 37 | 79 | - | - | 72 | - | - | 69 | - | - | 73 | - | - | 46 | - | - | 43 | - | - | 65 | - | - | 45 | - | - |
| 38 | 110 | - | - | 112 | - | - | 100 | - | - | 87 | - | - | 79 | - | - | 84 | - | - | 93 | - | - | 91 | - | - |
| 39 | 79 | - | - | 64 | - | - | 43 | - | - | 57 | - | - | 49 | - | - | 32 | - | - | 31 | - | - | 35 | - | - |
| 40 | 392 | - | - | 432 | - | - | 405 | - | - | 345 | - | - | 316 | - | - | 230 | - | - | 253 | - | - | 226 | - | - |
| 41 | 54 | - | - | 66 | - | - | 78 | - | - | 51 | - | - | 59 | - | - | 55 | - | - | 38 | - | - | 23 | - | - |
| 42 | 18 | - | - | 57 | - | - | 25 | - | - | 40 | - | - | 63 | - | - | 58 | - | - | 50 | - | - | 53 | - | - |
| 43 | 86 | - | - | 53 | - | - | 72 | - | - | 71 | - | - | 87 | - | - | 86 | - | - | 80 | - | - | 78 | - | - |
| 44 | 72 | - | - | 83 | - | - | 111 | - | - | 126 | - | - | 118 | - | - | 104 | - | - | 122 | - | - | 109 | - | - |
| 45 | 54 | - | - | 94 | - | - | 80 | - | - | 73 | - | - | 75 | - | - | 48 | - | - | 45 | - | - | 35 | - | - |
| 46 | 68 | - | - | 86 | - | - | 97 | - | - | 149 | - | - | 104 | - | - | 108 | - | - | 123 | - | - | 165 | - | - |
| 47 | 170 | - | - | 170 | - | - | 123 | - | - | 112 | - | - | 118 | - | - | 110 | - | - | 137 | - | - | 119 | - | - |

"-" indicates the value of <10.

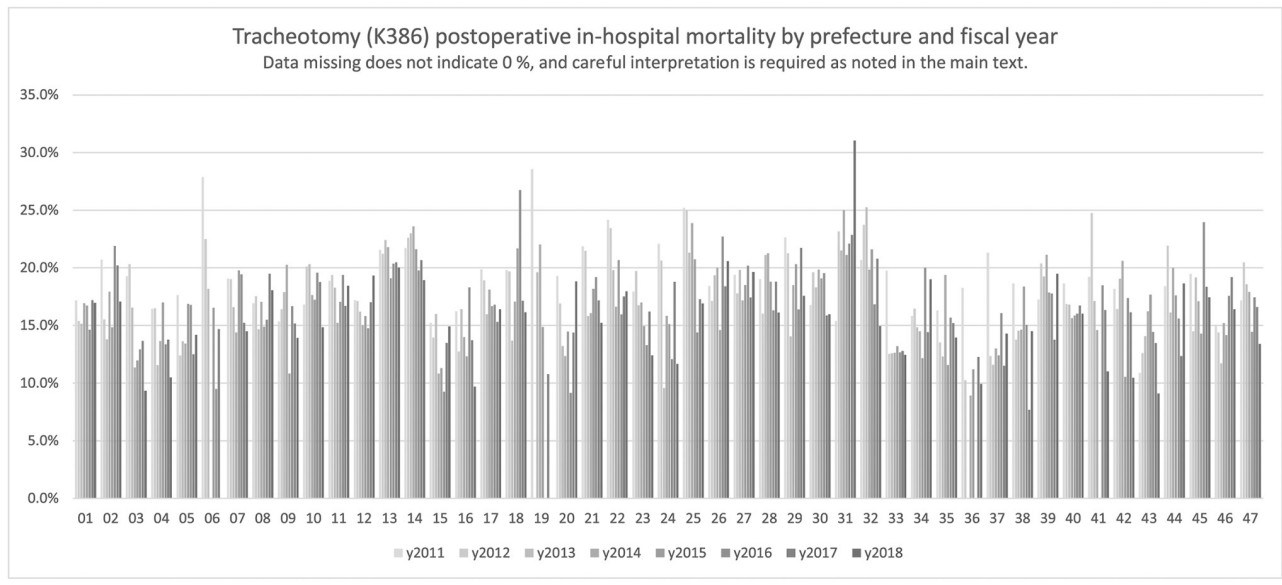

**Fig 4. Tracheotomy (K386).** Tracheotomy (K386) postoperative in-hospital mortality by prefecture and fiscal year.

future studies for information, such as knowledge about the fields requiring a detailed analysis or regions from where data ought to be collected and analyzed more thoroughly.

In the present study, in-hospital mortality was selected as the outcome; moreover, postoperative 30-day in-hospital mortality was considered. However, the number of outcomes within 30 days was naturally smaller and could increase missing data. Furthermore, being a rule of data publication, it was necessary to avoid inference from other data with cells having <10 records. For example, if presenting both in-hospital mortality and postoperative 30-day in-hospital mortality, it could be partly possible to calculate a small number of in-hospital mortality after 31 days. Therefore, only one type of outcome was used in this study; however, balancing the presentation of useful data with privacy considerations was challenging.

The resulting unattained numerical frames were rather expected; however, slightly more missing data were generated. For example, in coronary artery bypass surgery, the author had assumed that a large number of outcomes could have appeared and that those actual comparisons would be possible in more regions. However, both the number of outcomes and the number of cases were small in some regions; therefore, several numerical results were unobtainable. To circumvent this, it was possible to combine four separate coronary artery bypass procedures into one analysis or to integrate several similar procedures. It would be more useful to present both the combined and fine-grained aggregate values; however, in principle, we must avoid using those aggregate values to calculate a small numerical result. Thus, the more detailed results were presented in this study.

Deeper analysis is required to interpret the results associated with the missing values. Missing data should not be assumed to be zero or low values. Specifically, for comparison using graphs, the missing values appear to be zero because they do not exist or may get neglected. The missing values should be carefully re-evaluated differently, such as by integrating several fields, reanalyzing them, or referring to other information.

Furthermore, mortality outcomes are not necessarily bad outcomes. Analyses that use postoperative death or survival as an outcome can be primarily those that equate them with the "successfulness of surgery or perioperative procedures" to some extent. For example,

**Table 7. Tracheotomy (K386): Case number and outcomes of postoperative in-hospital mortality.**

| Fiscal year | 2011 | | | 2012 | | | 2013 | | | 2014 | | | 2015 | | | 2016 | | | 2017 | | | 2018 | | |
|---|---|---|---|---|---|---|---|---|---|---|---|---|---|---|---|---|---|---|---|---|---|---|---|---|
| Prefecture | Case | Outcome | | Case | Outcome | | Case | Outcome | | Case | Outcome | | Case | Outcome | | Case | Outcome | | Case | Outcome | | Case | Outcome | |
| 01 | 862 | 250 | 29.0% | 839 | 246 | 29.3% | 852 | 264 | 31.0% | 786 | 224 | 28.5% | 903 | 295 | 32.7% | 779 | 211 | 27.1% | 820 | 243 | 29.6% | 837 | 243 | 29.0% |
| 02 | 232 | 69 | 29.7% | 219 | 57 | 26.0% | 203 | 44 | 21.7% | 212 | 59 | 27.8% | 229 | 63 | 27.5% | 210 | 75 | 35.7% | 203 | 62 | 30.5% | 211 | 58 | 27.5% |
| 03 | 140 | 47 | 33.6% | 133 | 46 | 34.6% | 133 | 41 | 30.8% | 141 | 31 | 22.0% | 117 | 35 | 29.9% | 116 | 32 | 27.6% | 117 | 33 | 28.2% | 118 | 29 | 24.6% |
| 04 | 213 | 56 | 26.3% | 194 | 53 | 27.3% | 199 | 48 | 24.1% | 205 | 47 | 22.9% | 200 | 50 | 25.0% | 232 | 52 | 22.4% | 196 | 43 | 21.9% | 200 | 43 | 21.5% |
| 05 | 170 | 62 | 36.5% | 129 | 32 | 24.8% | 154 | 50 | 32.5% | 134 | 35 | 26.1% | 160 | 50 | 31.3% | 149 | 48 | 32.2% | 160 | 50 | 31.3% | 155 | 41 | 26.5% |
| 06 | 122 | 57 | 46.7% | 120 | 44 | 36.7% | 99 | 36 | 36.4% | 115 | 28 | 24.3% | 121 | 33 | 27.3% | 116 | 27 | 23.3% | 109 | 34 | 31.2% | 105 | 26 | 24.8% |
| 07 | 299 | 116 | 38.8% | 268 | 103 | 38.4% | 217 | 70 | 32.3% | 285 | 94 | 33.0% | 248 | 91 | 36.7% | 247 | 81 | 32.8% | 276 | 81 | 29.3% | 221 | 73 | 33.0% |
| 08 | 313 | 105 | 33.5% | 262 | 107 | 40.8% | 266 | 83 | 31.2% | 264 | 86 | 32.6% | 289 | 94 | 32.5% | 297 | 95 | 32.0% | 277 | 95 | 34.3% | 310 | 98 | 31.6% |
| 09 | 241 | 67 | 27.8% | 250 | 74 | 29.6% | 229 | 69 | 30.1% | 247 | 76 | 30.8% | 277 | 57 | 20.6% | 258 | 60 | 23.3% | 244 | 57 | 23.4% | 237 | 54 | 22.8% |
| 10 | 214 | 73 | 34.1% | 204 | 81 | 39.7% | 197 | 69 | 35.0% | 187 | 60 | 32.1% | 238 | 91 | 38.2% | 276 | 108 | 39.1% | 197 | 69 | 35.0% | 209 | 61 | 29.2% |
| 11 | 667 | 224 | 33.6% | 691 | 232 | 33.6% | 624 | 198 | 31.7% | 631 | 184 | 29.2% | 680 | 199 | 29.3% | 727 | 235 | 32.3% | 737 | 200 | 27.1% | 775 | 238 | 30.7% |
| 12 | 640 | 196 | 30.6% | 613 | 195 | 31.8% | 599 | 177 | 29.5% | 580 | 156 | 26.9% | 683 | 196 | 28.7% | 664 | 195 | 29.4% | 670 | 197 | 29.4% | 688 | 214 | 31.1% |
| 13 | 1917 | 620 | 32.3% | 1815 | 560 | 30.9% | 1740 | 558 | 32.1% | 1858 | 557 | 30.0% | 1921 | 550 | 28.6% | 1764 | 496 | 28.1% | 1784 | 510 | 28.6% | 1698 | 480 | 28.3% |
| 14 | 995 | 358 | 36.0% | 863 | 314 | 36.4% | 935 | 324 | 34.7% | 970 | 349 | 36.0% | 1004 | 304 | 30.3% | 982 | 292 | 29.7% | 1036 | 328 | 31.7% | 1014 | 267 | 26.3% |
| 15 | 184 | 57 | 31.0% | 172 | 57 | 33.1% | 175 | 46 | 26.3% | 157 | 40 | 25.5% | 168 | 44 | 26.2% | 162 | 49 | 30.2% | 163 | 49 | 30.1% | 181 | 50 | 27.6% |
| 16 | 154 | 49 | 31.8% | 157 | 46 | 29.3% | 134 | 35 | 26.1% | 150 | 43 | 28.7% | 138 | 36 | 26.1% | 153 | 50 | 32.7% | 153 | 39 | 25.5% | 124 | 34 | 27.4% |
| 17 | 136 | 50 | 36.8% | 111 | 40 | 36.0% | 119 | 42 | 35.3% | 138 | 49 | 35.5% | 120 | 50 | 41.7% | 125 | 44 | 35.2% | 111 | 46 | 41.4% | 128 | 50 | 39.1% |
| 18 | 101 | 37 | 36.6% | 61 | 25 | 41.0% | 95 | 32 | 33.7% | 82 | 28 | 34.1% | 83 | 32 | 38.6% | 86 | 37 | 43.0% | 70 | 21 | 30.0% | 62 | 22 | 35.5% |
| 19 | 56 | 22 | 39.3% | 57 | 15 | 26.3% | 51 | 13 | 25.5% | 59 | 21 | 35.6% | 74 | 27 | 36.5% | 64 | 15 | 23.4% | 102 | 25 | 24.5% | 76 | 20 | 26.3% |
| 20 | 228 | 79 | 34.6% | 213 | 64 | 30.0% | 189 | 64 | 33.9% | 162 | 45 | 27.8% | 166 | 47 | 28.3% | 153 | 36 | 23.5% | 167 | 52 | 31.1% | 154 | 57 | 37.0% |
| 21 | 215 | 93 | 43.3% | 177 | 64 | 36.2% | 215 | 71 | 33.0% | 193 | 63 | 32.6% | 187 | 65 | 34.8% | 177 | 62 | 35.0% | 198 | 68 | 34.3% | 184 | 58 | 31.5% |
| 22 | 393 | 142 | 36.1% | 371 | 144 | 38.8% | 379 | 139 | 36.7% | 343 | 107 | 31.2% | 329 | 108 | 32.8% | 301 | 82 | 27.2% | 314 | 89 | 28.3% | 334 | 99 | 29.6% |
| 23 | 596 | 201 | 33.7% | 604 | 197 | 32.6% | 645 | 176 | 27.3% | 653 | 194 | 29.7% | 650 | 182 | 28.0% | 669 | 150 | 22.4% | 685 | 204 | 29.8% | 644 | 156 | 24.2% |
| 24 | 163 | 59 | 36.2% | 160 | 54 | 33.8% | 115 | 25 | 21.7% | 158 | 47 | 29.7% | 172 | 52 | 30.2% | 174 | 52 | 29.9% | 165 | 50 | 30.3% | 163 | 40 | 24.5% |
| 25 | 131 | 48 | 36.6% | 144 | 59 | 41.0% | 122 | 43 | 35.2% | 134 | 53 | 39.6% | 135 | 52 | 38.5% | 139 | 43 | 30.9% | 133 | 50 | 37.6% | 142 | 51 | 35.9% |
| 26 | 331 | 122 | 36.9% | 362 | 112 | 30.9% | 310 | 103 | 33.2% | 335 | 108 | 32.2% | 336 | 99 | 29.5% | 361 | 129 | 35.7% | 337 | 107 | 31.8% | 345 | 117 | 33.9% |
| 27 | 1181 | 393 | 33.3% | 1141 | 375 | 32.9% | 1182 | 381 | 32.2% | 1135 | 337 | 29.7% | 1237 | 376 | 30.4% | 1234 | 396 | 32.1% | 1222 | 362 | 29.6% | 1293 | 383 | 29.6% |
| 28 | 683 | 219 | 32.1% | 674 | 201 | 29.8% | 597 | 198 | 33.2% | 602 | 200 | 33.2% | 675 | 214 | 31.7% | 674 | 222 | 32.9% | 670 | 215 | 32.1% | 633 | 176 | 27.8% |
| 29 | 243 | 93 | 38.3% | 207 | 66 | 31.9% | 199 | 54 | 27.1% | 227 | 63 | 27.8% | 202 | 69 | 34.2% | 244 | 70 | 28.7% | 221 | 66 | 29.9% | 222 | 76 | 34.2% |
| 30 | 167 | 46 | 27.5% | 153 | 49 | 32.0% | 142 | 40 | 28.2% | 121 | 40 | 33.1% | 131 | 38 | 29.0% | 133 | 39 | 29.3% | 126 | 41 | 32.5% | 119 | 32 | 26.9% |
| 31 | 91 | 29 | 31.9% | 95 | 41 | 43.2% | 93 | 34 | 36.6% | 84 | 32 | 38.1% | 90 | 31 | 34.4% | 86 | 24 | 27.9% | 105 | 40 | 38.1% | 116 | 56 | 48.3% |
| 32 | 121 | 40 | 33.1% | 118 | 57 | 48.3% | 103 | 41 | 39.8% | 121 | 52 | 43.0% | 125 | 66 | 52.8% | 101 | 50 | 49.5% | 101 | 43 | 42.6% | 87 | 35 | 40.2% |
| 33 | 309 | 106 | 34.3% | 287 | 80 | 27.9% | 262 | 83 | 31.7% | 277 | 82 | 29.6% | 280 | 88 | 31.4% | 292 | 74 | 25.3% | 297 | 85 | 28.6% | 305 | 82 | 26.9% |
| 34 | 322 | 82 | 25.5% | 334 | 110 | 32.9% | 310 | 88 | 28.4% | 290 | 83 | 28.6% | 304 | 81 | 26.6% | 305 | 100 | 32.8% | 333 | 107 | 32.1% | 305 | 101 | 33.1% |
| 35 | 221 | 72 | 32.6% | 207 | 56 | 27.1% | 195 | 47 | 24.1% | 227 | 89 | 39.2% | 190 | 56 | 29.5% | 249 | 81 | 32.5% | 204 | 78 | 38.2% | 215 | 69 | 32.1% |
| 36 | 115 | 26 | 22.6% | 127 | 20 | 15.7% | 120 | 18 | 15.0% | 112 | 23 | 20.5% | 125 | 32 | 25.6% | 98 | 14 | 14.3% | 106 | 24 | 22.6% | 121 | 30 | 24.8% |
| 37 | 169 | 60 | 35.5% | 170 | 38 | 22.4% | 138 | 36 | 26.1% | 146 | 35 | 24.0% | 153 | 39 | 25.5% | 137 | 45 | 32.8% | 113 | 33 | 29.2% | 126 | 30 | 23.8% |
| 38 | 134 | 43 | 32.1% | 138 | 43 | 31.2% | 117 | 43 | 36.8% | 123 | 32 | 26.0% | 136 | 53 | 39.0% | 133 | 42 | 31.6% | 143 | 38 | 26.6% | 138 | 30 | 21.7% |
| 39 | 174 | 60 | 34.5% | 157 | 56 | 35.7% | 135 | 47 | 34.8% | 142 | 55 | 38.7% | 168 | 64 | 38.1% | 135 | 45 | 33.3% | 138 | 40 | 29.0% | 118 | 43 | 36.4% |

*(Continued)*

**Table 7.** (Continued)

| Fiscal year | 2011 | | | 2012 | | | 2013 | | | 2014 | | | 2015 | | | 2016 | | | 2017 | | | 2018 | | |
|---|---|---|---|---|---|---|---|---|---|---|---|---|---|---|---|---|---|---|---|---|---|---|---|
| Prefecture | Case | Outcome | | Case | Outcome | | Case | Outcome | | Case | Outcome | | Case | Outcome | | Case | Outcome | | Case | Outcome | | Case | Outcome | |
| 40 | 826 | 263 | 31.8% | 807 | 257 | 31.8% | 720 | 221 | 30.7% | 755 | 219 | 29.0% | 794 | 216 | 27.2% | 799 | 213 | 26.7% | 843 | 262 | 31.1% | 817 | 227 | 27.8% |
| 41 | 104 | 34 | 32.7% | 101 | 35 | 34.7% | 111 | 34 | 30.6% | 96 | 26 | 27.1% | 105 | 26 | 24.8% | 119 | 38 | 31.9% | 98 | 27 | 27.6% | 109 | 25 | 22.9% |
| 42 | 220 | 78 | 35.5% | 201 | 58 | 28.9% | 194 | 62 | 32.0% | 170 | 55 | 32.4% | 199 | 50 | 25.1% | 190 | 59 | 31.1% | 186 | 55 | 29.6% | 172 | 42 | 24.4% |
| 43 | 257 | 56 | 21.8% | 214 | 48 | 22.4% | 220 | 54 | 24.5% | 228 | 59 | 25.9% | 232 | 79 | 34.1% | 201 | 51 | 25.4% | 193 | 48 | 24.9% | 209 | 39 | 18.7% |
| 44 | 201 | 64 | 31.8% | 187 | 65 | 34.8% | 180 | 69 | 38.3% | 200 | 77 | 38.5% | 193 | 64 | 33.2% | 186 | 64 | 34.4% | 170 | 48 | 28.2% | 177 | 67 | 37.9% |
| 45 | 154 | 56 | 36.4% | 145 | 48 | 33.1% | 146 | 45 | 30.8% | 152 | 45 | 29.6% | 161 | 39 | 24.2% | 146 | 54 | 37.0% | 147 | 43 | 29.3% | 149 | 41 | 27.5% |
| 46 | 279 | 88 | 31.5% | 264 | 94 | 35.6% | 239 | 87 | 36.4% | 204 | 69 | 33.8% | 247 | 93 | 37.7% | 239 | 91 | 38.1% | 255 | 99 | 38.8% | 262 | 96 | 36.6% |
| 47 | 355 | 112 | 31.5% | 342 | 107 | 31.3% | 269 | 83 | 30.9% | 268 | 78 | 29.1% | 256 | 65 | 25.4% | 321 | 90 | 28.0% | 265 | 75 | 28.3% | 276 | 68 | 24.6% |

"–" indicates the value of <10.

tracheostomies are often performed to continue the treatment of patients with respiratory disorders. In such cases, the result refers to whether death or survival can often be independent from surgical techniques and perioperative management. This can be true for several other surgeries. For example, a patient who would otherwise have died if a certain surgery, such as coronary artery bypass surgery, was not performed would die as a result of the procedure.

Nevertheless, comparisons of outcomes can and should be discussed from various perspectives. A perspective to consider is whether a surgery should have been performed even if there were "no" other possible treatments. Meanwhile, there may exist areas with inadequate medical care that cannot be implemented until the last resort. Therefore, regarding outcomes in quality of care, the background information of cases such as instances of case-mix must be considered. Such a detailed study interpretation requires a comprehensive analysis with a more elaborate protocol; however, it is important to first present the current status, such as in this study, and to keep the interpretation flexible.

Other minor details include normalizing the minor inconsistencies in the data. For example, the K-code may contain a 2-byte character "K" that is different from the original code or the data may be recorded with codes that do not exist. Therefore, the conversion and unification of these characters can be considered when using the dataset. Although "blood transfusion" was found in data that should not be reported as surgeries in the Form 1 of DPC data as per input rules, unlike pre-maintained databases such as registry databases, caution should be exercised while using unclean raw data. The present study requested data acquisition according to the standard established rules at the beginning of the study, and managing the data inconsistencies and small numerical cells as described above had to be planned. Subsequently, it is now possible to conduct the study using individual data, which should facilitate a more flexible analysis in the future.

## 5. Conclusions

The present study analyzed surgical outcomes at hospitals throughout Japan by region and over time using the highly comprehensive DPC database, provided comprehensive result data, and depicted regional differences and temporal changes.

## Supporting information

**S1 Data. All data obtained in this study.** All data obtained and all results in this study.
(XLSX)

## Author Contributions

**Conceptualization:** Susumu Kunisawa.

**Formal analysis:** Susumu Kunisawa.

**Funding acquisition:** Susumu Kunisawa.

**Investigation:** Susumu Kunisawa.

**Writing – original draft:** Susumu Kunisawa.

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
