## [Decision Letter · Decision Letter 0]

27 Mar 2023

PONE-D-23-01751Postoperative mortality analysis on nationwide data from diagnosis procedure combination database in JapanPLOS ONE

Dear Dr. Kunisawa,

Thank you for submitting your manuscript to PLOS ONE. After careful consideration, we feel that it has merit but does not fully meet PLOS ONE’s publication criteria as it currently stands. Therefore, we invite you to submit a revised version of the manuscript that addresses the points raised during the review process.

We look forward to receiving your revised manuscript.

Kind regards,

Lisa Kawatsu, PhD

Academic Editor

PLOS ONE

“Funding: This study was supported by Japan Society for the Promotion of Science (grant number 20K18961). The funders had no role in study design, data collection and analysis, decision to publish, or preparation of the manuscript.”

“This study was supported by Japan Society for the Promotion of Science (grant number 20K18961). The funders had no role in study design, data collection and analysis, decision to publish, or preparation of the manuscript.”

Additional Editor Comments:

Though it addresses an important issue, it is not clear how much the manuscript will interest international readers. As the Review 2 pointed out, the authors need to restructure the background and discussion sections, so as to make the manuscript more relevant not just in Japan but globally. It is also recommended that the authors re-consider about presentation of results. The tables simply filled with numbers are not easy for readers to capture what the main results are. Presenting the results in graphic figures may be one option, with raw numbers as supplementary data.

Reviewers' comments:

Reviewer's Responses to Questions

**Comments to the Author**

1. Is the manuscript technically sound, and do the data support the conclusions?

Reviewer #1: Yes

Reviewer #2: Yes

2. Has the statistical analysis been performed appropriately and rigorously? 

Reviewer #1: I Don't Know

Reviewer #2: Yes

3. Have the authors made all data underlying the findings in their manuscript fully available?

Reviewer #1: Yes

Reviewer #2: Yes

4. Is the manuscript presented in an intelligible fashion and written in standard English?

Reviewer #1: Yes

Reviewer #2: Yes

5. Review Comments to the Author

Reviewer #1: The study has an impact on the practice to improve clinical outcomes in Japanese patients undergoing surgeries. It is of clinical significance although careful considerations to interpret the results are needed.

Reviewer #2: The authors reported on graphical and temporal changes in mortality rates surgical procedures using nationwide database. As the results may be challenging for readers outside Japan to understand, this reviewer suggests that the authors describe their findings from a more general point of view.

In the introduction section, the authors focused primarily on the National Clinical Database and diagnosis procedure combination database, which are both nationwide domestic database in Japan. However, it is recommended to begin with more general statements that emphasize the limitation not only of domestic datasets but also of global datasets and the unique contributions of the current analysis.

In regards to the material and methods section, it is less meaningful to include database-specific terms such as “K-code”, “-“, and “*-“ in the section. Instead, it is recommended to use more commonly understood terms. Additionally, it is important to state the reasons why values of < 10 are not analyzed (for the anonymization purposes?).

In the methods section, it is not clear why the authors focused on four specific procedures (K0811, K1771, K5521-552-22, and K386) in the results section. The discussion section suggests that regional differences and temporal changes can be observed to some extent in the complete results, but this information is not provided in the current manuscript.

In the opinion of this reviewer, the easiest way to avoid inference from other data with cells having <10 records is not to split data into 47 prefectures but rather to split data into larger area such as Kanto, Kinki etc. Please clearly state the rationale to analyze the dataset at the prefectural level.

6. PLOS authors have the option to publish the peer review history of their article (what does this mean?). If published, this will include your full peer review and any attached files.

Reviewer #1: **Yes: **Hideki ISHII

Reviewer #2: **Yes: **Kyohei Yamaji

---

## [Author Response · Author response to Decision Letter 0]

6 Apr 2023

For Journal requirements:

Thank you for the notifications. I have removed that part from my manuscript, and I have no update for the funding information. 

I have added the ethics statement in the last paragraph of the Methods section.

For Additional Editor Comments:

Thank you for your advice. I have revised the Background and Discussion sections for providing a global perspective.

For Reviewer #1:

Thank you for your careful review.

For Reviewer #2:

Thank you very much for your kind advice.

I agree that the graphs help in clearly explaining the data. Considering that graphs were the main outputs of this study, I have revised the manuscript so that readers can understand the graphs easily.

I have revised the Discussion section for providing a general perspective.

 I have revised the Introduction section for providing a more global perspective.

It is true that K-code is unique to Japan, and foreign readers find it difficult to understand this code. However, since the actual data were coded using this system and the Supporting Information provided the original data, I have revised the text and stated that this code is unique to Japan. I have also emphasized on the importance of presenting this code in this report.

Regarding “-“ and “*-“, I set these symbols originally when I deleted the data to accommodate the minimum number. I have revised the text to explain the significance of these symbols clearly.

Anonymization rule was set by the government to enhance anonymity. I have added this information in the revised manuscript.

In Material and methods, I had described the selection criteria; I have revised the text describing the criteria to enhance clarity.

I agree with your advice. Originally, I wanted to look at the differences in a relatively more detailed regional units; for example, at the level of secondary medical regions or cities. However, I thought that this would reduce the number of nulls, so I planned to work on the prefectural level, which is one of the general units. I thought that it would be good grouping to some extent. The results showed fewer outcomes in particular and more nulls than I expected.

I revised the text to indicate this more clearly.

Additionally, I have mentioned in the last paragraph of the Discussion section that this report follows the standard rules. However, the rule has now changed, and it is now possible to conduct the study using individual data, which could help in performing a flexible analysis in the future, such as by changing the grouping to the proper levels.

---

## [Decision Letter · Decision Letter 1]

12 May 2023

Postoperative mortality analysis on nationwide data from diagnosis procedure combination database in Japan

PONE-D-23-01751R1

Dear Dr. Kunisawa,

We’re pleased to inform you that your manuscript has been judged scientifically suitable for publication and will be formally accepted for publication once it meets all outstanding technical requirements.

Kind regards,

Lisa Kawatsu, PhD

Academic Editor

PLOS ONE

Additional Editor Comments (optional):

Reviewers' comments:

Reviewer's Responses to Questions

**Comments to the Author**

1. If the authors have adequately addressed your comments raised in a previous round of review and you feel that this manuscript is now acceptable for publication, you may indicate that here to bypass the “Comments to the Author” section, enter your conflict of interest statement in the “Confidential to Editor” section, and submit your "Accept" recommendation.

Reviewer #1: All comments have been addressed

Reviewer #2: All comments have been addressed

2. Is the manuscript technically sound, and do the data support the conclusions?

Reviewer #1: Yes

Reviewer #2: Yes

3. Has the statistical analysis been performed appropriately and rigorously? 

Reviewer #1: Yes

Reviewer #2: Yes

4. Have the authors made all data underlying the findings in their manuscript fully available?

Reviewer #1: Yes

Reviewer #2: Yes

5. Is the manuscript presented in an intelligible fashion and written in standard English?

Reviewer #1: Yes

Reviewer #2: Yes

6. Review Comments to the Author

Reviewer #1: (No Response)

Reviewer #2: The authors addressed all the comments from this reviewer. No additional comments from this reviewer.

7. PLOS authors have the option to publish the peer review history of their article (what does this mean?). If published, this will include your full peer review and any attached files.

Reviewer #1: **Yes: **Hideki ISHII

Reviewer #2: No

---

## [Editor Report · Acceptance letter]

30 May 2023

PONE-D-23-01751R1 

Postoperative mortality analysis on nationwide data from diagnosis procedure combination database in Japan 

Dear Dr. Kunisawa:

I'm pleased to inform you that your manuscript has been deemed suitable for publication in PLOS ONE. Congratulations! Your manuscript is now with our production department. 

Kind regards, 

on behalf of

Dr. Lisa Kawatsu 

Academic Editor

PLOS ONE